# Neural Slot Interpreters: Grounding Object Semantics in Emergent Slot Representations

**Bhishma Dedhia**      **Niraj K Jha**
*Department of Electrical and Computer Engineering*
*Princeton University*
*{ bdedhia, jha}@princeton.edu*

**Reviewed on OpenReview:** *https://openreview.net/forum?id=lyxRBPmmnV*

## Abstract

Several accounts of human cognition posit that our intelligence is rooted in our ability to *form* abstract composable concepts, *ground* them in our environment, and *reason* over these grounded entities. This trifecta of human thought has remained elusive in modern intelligent machines. In this work, we investigate whether slot representations extracted from visual scenes serve as appropriate compositional abstractions for grounding and reasoning. We present the Neural Slot Interpreter (NSI), which learns to ground object semantics in slots. At the core of NSI is a nested schema that uses simple syntax rules to organize the object semantics of a scene into object-centric schema primitives. Then, the NSI metric learns to ground primitives into slots through a structured contrastive learning objective that reasons over the intermodal alignment. Experiments with a bi-modal object-property and scene retrieval task demonstrate the grounding efficacy and interpretability of correspondences learned by NSI. From a scene representation standpoint, we find that emergent NSI slots that move beyond the image grid by binding to spatial objects facilitate improved visual grounding compared to conventional bounding-box-based approaches. From a data efficiency standpoint, we empirically validate that NSI learns more generalizable representations from a fixed amount of annotation data than the traditional approach. We also show that the grounded slots surpass unsupervised slots in real-world object discovery and scale with scene complexity. Finally, we investigate the downstream efficacy of the grounded slots. Vision Transformers trained on grounding-aware NSI tokenizers using as few as ten tokens outperform patch-based tokens on challenging few-shot classification tasks.

## 1 Introduction

Humans possess a repertoire of strong structural biases, a kind of abstract knowledge that enables us to perceive and rapidly adapt to our environments (Griffiths et al., 2010). *Compositionality* is one such structural prior that helps us systematically reason about complex stimuli as a whole by recursively reasoning about its parts (Zuberbühler, 2019; Lake & Baroni, 2023). We decompose broad motor skills into finer dexterous finger movements, sentences into words and phrases, and speech into phonemes. In the visual world, the concept of "objectness" serves as a natural compositional prior, enabling us to decompose novel scenes into familiar objects and reason about their properties (Lake et al., 2016). We also have the uncanny ability to connect real-world entities and concepts to these abstract object-like symbols in our heads, canonically referred to as the *grounding problem* (Harnad, 1990; Greff et al., 2020). For instance, human infants, while looking at a zebra for the first time, might excitedly conclude that it is, in fact, "a striped horse." *If grounded object-like representations are fundamental to human-like compositional generalization, how do we instill these inductive biases into neural network representations*?

Unsupervised object-centric autoencoder models (Burgess et al., 2019; Greff et al., 2019; 2020; Locatello et al., 2020; Engelcke et al., 2019; 2021; Singh et al., 2021; Chang et al., 2023b; Seitzer et al., 2023; Kori

et al., 2024a) have become increasingly adept at learning object-centric representations called *slots* from raw visual stimuli. Further work has demonstrated that learned slots can be flexibly composed together for tasks like scene composition, causal induction, learning intuitive physics, dynamics simulation, and control (Dedhia et al., 2023; Jiang et al., 2023; Wu et al., 2023a;b; Chang et al., 2023a; Jabri et al., 2023). While object slots hold promise as a compositional building block for machines that mimic human abstraction and generalization, a key challenge emerges. Unlike humans, these learned slots lack grounding in real-world concepts. For example, a slot representation of an object like "apple" could refer to the fruit, the company, or a generic round artifact. Without grounding, a slot-based system cannot disambiguate these meanings effectively (Haugeland, 1985) and is fundamentally limited in its embodied reasoning abilities. Then the central question that guides our work is:

*"Can slots serve as effective representations to ground object semantics in visual scenes?"*

Grounding, in the context of slot representations, refers to the ability to associate labels (for e.g., identifying objects and their properties) with slot representations learned from a scene's visual features. Prior works (Locatello et al., 2020; Seitzer et al., 2023; Kori et al., 2024a) have tackled learning to ground slots by predicting object properties (texture, material, category, etc.) from the representations. The grounding objective is, therefore, *implicit* within the prediction of object semantics. However, ground truth correspondences between object concepts and slots are generally unknown, restricting prediction to a set-matching template. Under a set-matching framework, a *single* slot predicts object properties of a *single* object, thereby constraining the grounding information assimilated per slot. We circumvent the limitations of prediction as a surrogate for grounding by making the grounding objective *explicit* in the form of a co-training paradigm that we call the *Neural Slot Interpreter* (NSI).

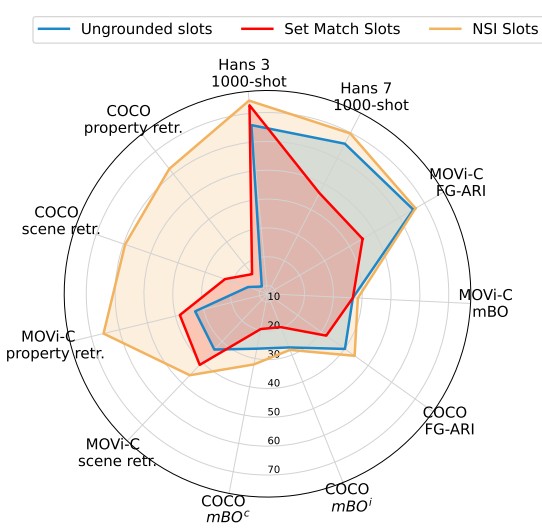

Figure 1: NSI abstracts grounded slots from scenes and enhances object discovery, grounding efficacy, and downstream reasoning abilities of slot representations.

The core insight behind NSI is simple: instead of *predicting* a single object-concept from a slot, *assign* multiple concepts to slots over a shared latent space. Our primary contribution is a similarity metric that explicitly reasons about the intermodal assignments. Notably, the proposed metric for NSI supplants the one-object-per-slot assumption and facilitates flexible assignment. Contrastive learning over the similarity objective yields grounded slots that outperform their ungrounded and set-matched counterparts over a broad swath of tasks (see Fig. 1). We propose an object-centric annotation schema in Section 4.1 for dense alignment to organize scene annotations for grounding into slots. We describe the design of a hierarchical transformer-based architecture in Section 4.2.2 to extract neural representations from the schema. To enable slots to ground a wide array of concepts flexibly without relying on matching templates, we formulate a bi-level scoring metric over a learned latent space in Section 4.2.

*Does NSI necessitate training slot-centric architectures from scratch?* Our experiments demonstrate that slots learned from pre-trained object-centric backbones (Seitzer et al., 2023) are easily adaptable to the NSI objective. *Are notions of objects effectively grounded in emergent slot representations?* We explicitly evaluate grounding efficacy through bi-modal retrieval tasks (Section 5.2.1), where models must retrieve corresponding object properties given a scene, and *vice versa*. Our experiments demonstrate that NSI outperforms approaches based on ungrounded slots, set-matching, and non-compositional embeddings. *How does NSI compare against traditional weakly supervised visual grounding?* In Section 5.2.2 we show that conventional bounding box abstractions are ineffective at grounding object semantics compared to NSI. *Can NSI be effective across different annotation size regimes?* We train NSI on different annotation levels and discuss it in Section 5.2.2. We find that NSI enables grounded slots to

be data-efficient, especially on real-world data. *Do slots that emerge from the NSI objective preserve object discovery abilities?* NSI slots preserve and often improve object discovery, as discussed in Section 5.3, where we demonstrate its competitiveness on object discovery benchmarks. *Can NSI-grounded slots be effective substrates for downstream tasks*? In Section 5.4, we discuss training of a vision transformer (ViT) (Dosovitskiy et al., 2021) for a scene classification task where significantly fewer grounded slot tokens show improved performance and adaptability over traditional patch-based tokens. *Can dense associations learned by NSI inform real-world reasoning systems*? Our experiments described in Appendix E.1 show the usefulness of learned correspondences in identifying and locating objects in diverse scenes.

Concretely, our contributions are as follows:

1. We present NSI (Section 4), a co-training grounding paradigm for object-centric learners. We use a nested object-centric annotation schema for dense slot-label alignment (Section 4.1). We formulate a similarity metric that measures scene-schema similarity by recursively reasoning over the similarity of compositional attributes of the respective modalities (Section 4.2.3). NSI utilizes the metric to ground slots via a contrastive learning objective (Section 4.2.4).

2. Our model demonstrates the effectiveness of NSI-trained slot representations at grounding object properties via a bimodal retrieval task (Section 5.2.1). NSI significantly improves retrieval performance compared to Hungarian Matching Criterion (HMC) matching slots, ungrounded slots, and non-compositional embedding baselines. Moreover, dense and interpretable correspondences between slot masks and object properties emerge from the NSI similarity metric.

3. We also compare NSI to conventional methods in weakly supervised grounding where practitioners traditionally utilize bounding box regions to encode visual object features (Section 5.2.2). Overlapping objects within coarser bounding box regions impede accurate object property assignment, a problem mitigated by learned slot masks via NSI. We also probe the data efficiency of both approaches and find that for a given set of annotations, NSI is more adept at aligning object semantics to scenes.

4. Our experiments also demonstrate that slots grounded via NSI improve slot performance over a wide array of tasks that encompass (1) object discovery (Section 5.3), (2) few-shot scene classification (Section 5.4), and (3) object detection (Appendix E.1). Overall, we find that grounded slot representations are key to object-centric perception, property grounding, and downstream adaptability for object-centric reasoning.

## 2 Related Work

**Object-Centric Learning.** Researchers have formulated inductive biases for learning composable visual representations called 'slots' from raw visual stimuli (Burgess et al., 2019; Greff et al., 2019; Locatello et al., 2020; Greff et al., 2020; Engelcke et al., 2019; 2021; Singh et al., 2021; Seitzer et al., 2023) and auxiliary temporal information (Kipf et al., 2021; Elsayed et al., 2022; Singh et al., 2022). While this line of work demonstrates unsupervised object discovery, the adoption of slot representations for grounding scenes remains largely underexplored. Prior works have been limited to using the HMC to align slots to ground-truth property labels for property prediction (Locatello et al., 2020) or fine-tuning shallow property predictors on pre-trained backbones (Seitzer et al., 2023). A recent work (Kori et al., 2024a) improves the predictive power of slots by learning quantized factorised priors to foster invariance of slots with object properties they represent. However, these methods, often relying on HMC for training or evaluation, fundamentally operate under a one-object-per-slot prediction constraint. This limits the richness of grounding information assimilated per slot and poses challenges in learning highly specialized representations, particularly in complex scenes. An orthogonal body of work aims to learn identifiable and interpretable slots without supervision. For instance, Kori et al. (2024b) focus on providing theoretical identifiability guarantees for unsupervised object-centric learning through probabilistic slot attention. Stanić et al. (2023) investigate synchrony-based models using complex-valued slot representations and contrastive learning for unsupervised object discovery. Similarly, Baldassarre & Azizpour (2022) incorporate contrastive losses to facilitate symmetry-breaking between slots. Furthermore, efforts like Zhang et al. (2023) focus on improving the core slot attention mechanism itself, using optimal transport to enhance tie-breaking in dynamic scenes. In contrast to these directions that focus

on unsupervised identifiability, our NSI method directly tackles the challenge of explicit semantic grounding through annotation-driven contrastive alignment. Moreover, intepretable slot features can potentially be integrated within the NSI framework to improve alignment.

**Visual Grounding.** Several bodies of works have grounded text into scenes by pre-training large-scale transformers using Internet-scale natural language supervision (Chen et al., 2020; Desai & Johnson, 2021; Li et al., 2019; 2020; Radford et al., 2021). Subsequent work has improved modeling of the multimodal streams by explicitly imbibing compositional synergy between visual concepts and text phrases. A prominent line of work trains predictive models on labeled text-bounding box correspondences (Chen et al., 2020; Kamath et al., 2021; Gan et al., 2020; Lu et al., 2019) but is expensive due to its fully supervised nature. To reduce the reliance on pre-specified correspondences for training, like NSI, several works propose weakly supervised visual grounding (Karpathy & Fei-Fei, 2015; Gupta et al., 2020; Wang et al., 2021a; Liu et al., 2024), but these approaches are still reliant on object detectors for bounding box proposals. Moreover, bounding boxes underspecify overlapping or occluded objects, thus making them a poor choice for grounding object semantics beyond categories. A recent work by Xu et al. (2022) goes beyond image patches to hierarchically discover semantic masks but relies on careful stacking of ViTs (Dosovitskiy et al., 2021) to achieve necessary object granularity. As opposed to completely neural methods, neuro-symbolic approaches use symbolic domain-specific languages to reason about image concepts (Johnson et al., 2017; Yi et al., 2018; Wang et al., 2021b). Such methods are extremely effective at question-answering but rely heavily on well-engineered logic priors and fail to generalize beyond synthetic data.

**Visual Tokenizers.** Patch-based tokens have been adopted as the standard for visual understanding (Dosovitskiy et al., 2021) and generation (Peebles & Xie, 2023). Variations of this template include discretized patch tokens (Du et al., 2024), mixed-resolution patch tokens (Ronen et al., 2023), and pruned patch tokens (Kong et al., 2022; Tang et al., 2023). Beyond patches, recent works have explored region-based tokens (Ma et al., 2024). However, these tokenizers are inherently grounding-agnostic, in contrast to humans, who possess the ability to abstract concepts based on linguistic or cultural grounding priors, (Segall et al., 1966; Winawer et al., 2007). To this end, our work explores a grounding-aware tokenizer.

## 3 Preliminaries

We detail a few essential preliminaries in this section. Readers should refer to the original manuscripts for further details.

### 3.1 Slot Attention

Object-centric learning frameworks decompose scenes by organizing them into compact representations called slots. Slot Attention (SA) (Locatello et al., 2020) is a powerful iterative attention mechanism for learning such slots from perceptual features extracted from vision backbones. At iteration $t$, for $L$ features $H \in \mathbb{R}^{L \times c}$ and $K$ slots $S^t \in \mathbb{R}^{K \times d}$, the slots compete to explain the features as follows:

$$M = \frac{K(H)Q(S^t)^T}{\sqrt{D}} \in \mathbb{R}^{L \times K}; A_{ij} = \frac{e^{M_{ij}}}{\sum_{j \in \{1, \cdots, K\}} e^{M_{ij}}} \quad (1)$$

$$\text{Update} = W^T V(S^t) \quad \text{where} \quad W_{ij} = \frac{A_{ij}}{\sum_{i \in \{1, \cdots, N\}} A_{ij}} \quad (2)$$

$D$ denotes the dimension of the attention head. Here, $Q(.) \in \mathbb{R}^{d \times D}$, $K(.) \in \mathbb{R}^{c \times D}$, $V(.) \in \mathbb{R}^{d \times D}$ are learned query, key, and value matrices, respectively. After attention-based weighted aggregation of features, a Gated Recurrent Unit $\mathcal{G}(.)$ (Cho et al., 2014) updates the slots as follows:

$$S^{t+1} = \mathcal{G}(\text{state} = S^t, \text{input} = \text{Update}) \quad (3)$$

The aggregation and updates are performed over multiple iterations to obtain the final representations. Slots have been empirically shown to bind to the features of the same object as the forward iteration proceeds, yielding per-object representations. We refer the reader to the original article by Locatello et al. (2020) to understand the complete rollout.

## 3.2 DINOSAUR Slot Attention

The original slot attention could learn slots on synthetic objects but failed to scale to real-world scenes. A recent object-centric learning method by Seitzer et al. (2023), called DINOSAUR, uses the DINO vision backbone (Caron et al., 2021) to scale slot representation learning to complex real-world scenes. The fundamental insight of the work lies in using features from strong, pre-trained encoders to perform slot attention instead of pixel-level reconstruction used by prior works. The authors use the rich representations from DINO as a substrate for learning grouped slots that capture spatial objectness features. Motivated by these findings, we adopt the pre-trained DINOSAUR as our slot-attention backbone for NSI and fine-tune it for semantic grounding.

## 3.3 Hungarian Matching Criterion (HMC) for Prediction using Slots

Learning object properties from slots is a challenging problem because the ground truth correspondences between objects and slots are unknown. Practitioners circumvent this problem by assuming that every slot contains a single object and minimize the costs associated with the bijection mapping from the set of slots to the set of objects using the Hungarian Algorithm (Kuhn, 1955). Given a set of slots $\mathcal{S}^{1:K}$, object properties $\mathcal{P}^{1:K}$, and assignment cost function $\mathcal{C}(.)$, the Hungarian algorithm is a polynomial-time algorithm that finds the optimal assignment $U^*(.)$:

$$U^* = \arg\min_U \sum_{i \in \{1, \cdots, K\}} \mathcal{C}\left(S^i, P^{U(i)}\right) \tag{4}$$

HMC is the primary method utilized by prior methods (Locatello et al., 2020; Seitzer et al., 2023; Kori et al., 2024a) to predict object properties from slots.

## 4 Neural Slot Interpreters

Recall that the goal of NSI is to ground concepts into slot representations such that the objects contained within the slot align with the embodied notions of the object. We begin by discussing the proposed organization for scene labels.

### 4.1 Nested Object-Centric Annotation Schema

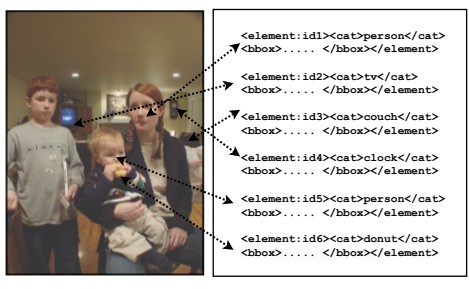

Figure 2: Description of a real-world scene using the nested schema. The dotted arrows show correspondences between primitives and the objects they annotate.

We propose a simple nested schema for expressing scene labels as a collection of objects and their associated properties. At a higher level, instances in the schema comprise multiple object-centric *primitives* (`<element>`). Primitives, in turn, contain respective object properties that form atomic units of the schema. Each primitive identifies a unique object in the scene. The nested properties `<p_j>` $(p_1, \cdots, p_J)$ form the children of the parent objects `<element>`. Some examples of `<p_j>` properties are, but not limited to, shape, material, category, and object position. Thus, instance primitives naturally capture the notion of an object, and neural representations extracted from primitives are, as such, well-suited for being grounded in slots. In Section 5.1, we demonstrate the straightforward application of the schema to organize labels on popular datasets. See Fig. 2 for an example instance and Appendix A.3 for more examples. On a more practical note, such schemas are commonly used software abstractions and can be ubiquitously interfaced with graphics engines, web APIs, or even large language models for semantic understanding (Dunn et al., 2022; Bubeck et al., 2023).

## 4.2 NSI Grounding Technique

Scenes and their corresponding schema instances capture object-centric representations through slots and primitives, respectively. NSI learns to align the object-centric representations of the respective modalities, i.e., slots and primitives, by grounding neural representations of schema primitives into slots (see Fig. 3). The grounding is learned by optimizing a contrastive learning objective over ground-truth scene-schema pairs (see Fig. 4). We describe the scene and schema encoder next.

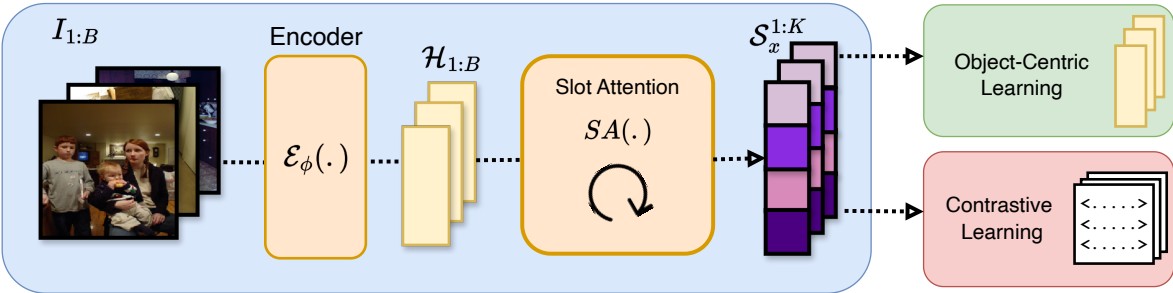

Figure 3: NSI overview. NSI augments object-centric learning autoencoders with a contrastive learning objective over a batch of scene-schema pairs. A DINOSAUR backbone (Seitzer et al., 2023) extracts slot representations $\mathcal{S}_x^{1:K}$ from a batch of scenes and a schema encoder extracts neural primitives $\mathcal{Z}_y^{1:N}$ from their corresponding schema pair. The slots are then passed to a decoder for reconstruction and the slot-primitive neural pairs are passed to the contrastive learning objective.

### 4.2.1 Learning Scene Representations

A given scene $I_x$ is represented via $K$ slots $S_x^{1:K} \in \mathbb{R}^{K \times d}$ abstracted from its perceptual features $H_x \in \mathbb{R}^{L \times c}$ (see Fig. 4 (a)) For a given feature extractor $\mathcal{E}_\phi(.)$, the slots are obtained as

$$H_x = \mathcal{E}_\phi\left(I_x\right) \quad ; \quad S_x^{1:K} = SA\left(H_x\right) \rightarrow \text{Slot Attention} \tag{5}$$

A spatial broadcast decoder $\mathcal{D}_\theta(.)$ (Locatello et al., 2020) reconstructs the features from slots, with the reconstruction error used as a learning signal:

$$\hat{H}_x = \mathcal{D}_\theta\left(S_x^{1:K}\right) \quad ; \quad \mathcal{L}_{recon} = \left\|H_x - \hat{H}_x\right\|^2 \tag{6}$$

### 4.2.2 Learning Schema Representations

A bi-level architecture (see Fig 4 (b)) learns neural representations of the schema primitives. First, a lower-level primitive encoder learns property-specific dictionaries $D(.)$ and embeds the property features into neural primitive representations. For discrete-valued properties, $D(.)$ is modeled as a simple lookup table of learnable weights, while continuous-valued properties are embedded via multi-layered perceptrons (MLPs). Let the dictionary $D_j(.)$ learn features for property $p_j$. Then, a primitive embedding $Z_{prim}$ is computed as:

$$Z = concat\left[D_1(p_1), \cdots, D_J(p_J)\right] \quad ; \quad Z_{prim} = MLP(Z) \tag{7}$$

Note that these lower-level representations are schema-agnostic and only capture object-specific features. Then an upper-level schema encoder uses a bidirectional schema Transformer to further embed primitives, endowing representations with the overall schema context. For a given schema instance $P_y$ with $N$ primitives, the final representations $Z_y^{1:N} \in \mathbb{R}^{N \times d}$ are computed via Transformer $\mathcal{T}_{schema}(.)$ as:

$$Z_y^{1:N} = \mathcal{T}_{schema}\left(Z_{prim}^1, \cdots, Z_{prim}^N\right) \tag{8}$$

### 4.2.3 Compositional Score Aggregation

Recall that we want to ground entities $Z_y^{1:N}$ into entities $S_x^{1:K}$. As a first step, we project these embeddings into a shared semantic space $Y \in \mathbb{R}^{d_{proj}}$. The projection head $\mathcal{H}_{scene}(.)$ for slots is modeled as the following residual network:

$$\tilde{Y}_x^k = W_{proj} S_x^k, \; W \in \mathbb{R}^{d_{proj} \times d} \quad ; \quad Y_x^k = \tilde{Y}_x^k + MLP\left(LayerNorm\left(\tilde{Y}_x^k\right)\right) \tag{9}$$

Here, $LayerNorm$ denotes the layer normalization operation. A separate residual head $\mathcal{H}_{schema}(.)$ projects the primitive representations $Z_y^{1:N}$ into the semantic embeddings $Y_y^{1:N}$. Next, we supplant the traditional single object per slot assumption by assigning each primitive to its nearest slot in the latent space, as measured by dot-product similarity. The similarity score $\mathcal{S}_{xy}$ between a scene $x$ and primitive $y$ is the sum of nearest-neighbor similarities resulting from the primitive-slot assignment (see Fig. 4 (c)).

$$k_n^* = \underset{k \in \{1, \cdots, K\}}{\arg \max} \; Y_x^{k^T} Y_y^n \quad ; \quad \mathcal{S}_{xy} = \sum_{n \in \{1, \cdots, N\}} \max_{k \in \{1, \cdots, K\}} \left(Y_x^{k^T} Y_y^n\right) \tag{10}$$

### 4.2.4 Contrastive Learning Objective

The modality-specific embeddings and the resultant grounding are learned by optimizing a contrastive learning objective (Fig 4 (d)). More precisely, given a $B$-sized batch of {scene, schema} pairs, we use the $\mathcal{S}_{xy}$ scores to distinguish the $B$ correct pairs from the $B^2 - B$ incorrect pairs. The probability of correctly classifying schema $P_x$ as the true pairing for scene $I_x$ (and conversely predicting $I_x$ from $P_x$) is formulated as follows:

$$\mathbb{P}_x^{schema} = \frac{exp\left(\mathcal{S}_{xx}/\tau\right)}{\sum_{y \in \{1, \cdots, B\}} exp\left(\mathcal{S}_{xy}/\tau\right)} \quad ; \quad \mathbb{P}_x^{scene} = \frac{exp\left(\mathcal{S}_{xx}/\tau\right)}{\sum_{y \in \{1, \cdots, B\}} exp\left(\mathcal{S}_{yx}/\tau\right)} \tag{11}$$

Here, the calculated scores are interpreted as logits and $\tau$ denotes the temperature parameter. The cross-entropy losses for scene and schema prediction are given by:

$$\mathcal{L}_{schema} = - \sum_{x \in \{1, \cdots, B\}} log\left(\mathbb{P}_x^{schema}\right) \quad ; \quad \mathcal{L}_{scene} = - \sum_{x \in \{1, \cdots, B\}} log\left(\mathbb{P}_x^{scene}\right) \tag{12}$$

The global contrastive learning objective is based on the InfoNCE loss (van den Oord et al., 2019) as follows:

$$\mathcal{L}_{contrastive} = (\mathcal{L}_{scene} + \mathcal{L}_{schema})/2 \tag{13}$$

The overall training objective for NSI is given by:

$$\mathcal{L}_{train} = \beta_1 \times \mathcal{L}_{contrastive} + \beta_2 \times \mathcal{L}_{recon} \tag{14}$$

Note that $\beta_1 = 0.0, \beta_2 = 1.0$ corresponds to traditional autoencoder object-centric learning frameworks. The learning objective can be interpreted as estimating a lower bound on the Mutual Information (MI) between the scene and schema distribution by optimizing the compatibility of their compositional embeddings. Let the MI between the joint scene and schema distribution $p(.)$ be defined as:

$$MI(I, P) = \mathbb{E}_{(I_x, P_y) \sim p(I, P)} \left[log \frac{p(I_x, P_y)}{p(I_x) p(P_y)}\right] \tag{15}$$

van den Oord et al. (2019) show that the optimized InfoNCE distributions $\mathbb{P}_{schema}^*$ and $\mathbb{P}_{scene}^*$ provide a low variance estimator of MI (which is generally intractable for high-dimensional data). The InfoNCE lower bound on MI is given by

$$MI(I, P) \geq log(B) - \mathcal{L}_{schema} \quad ; \quad MI(I, P) \geq log(B) - \mathcal{L}_{scene} \tag{16}$$

$$\Rightarrow MI(I, P) \geq log(B) - \frac{\mathcal{L}_{schema} + \mathcal{L}_{scene}}{2} \equiv log(B) - \mathcal{L}_{contrastive} \tag{17}$$

which becomes tighter as $B$ becomes larger. Therefore, minimizing the NSI contrastive loss maximizes a lower bound on MI. Algorithm 1 presents the NSI pseudocode for aggregating local alignments and computing the contrastive loss.

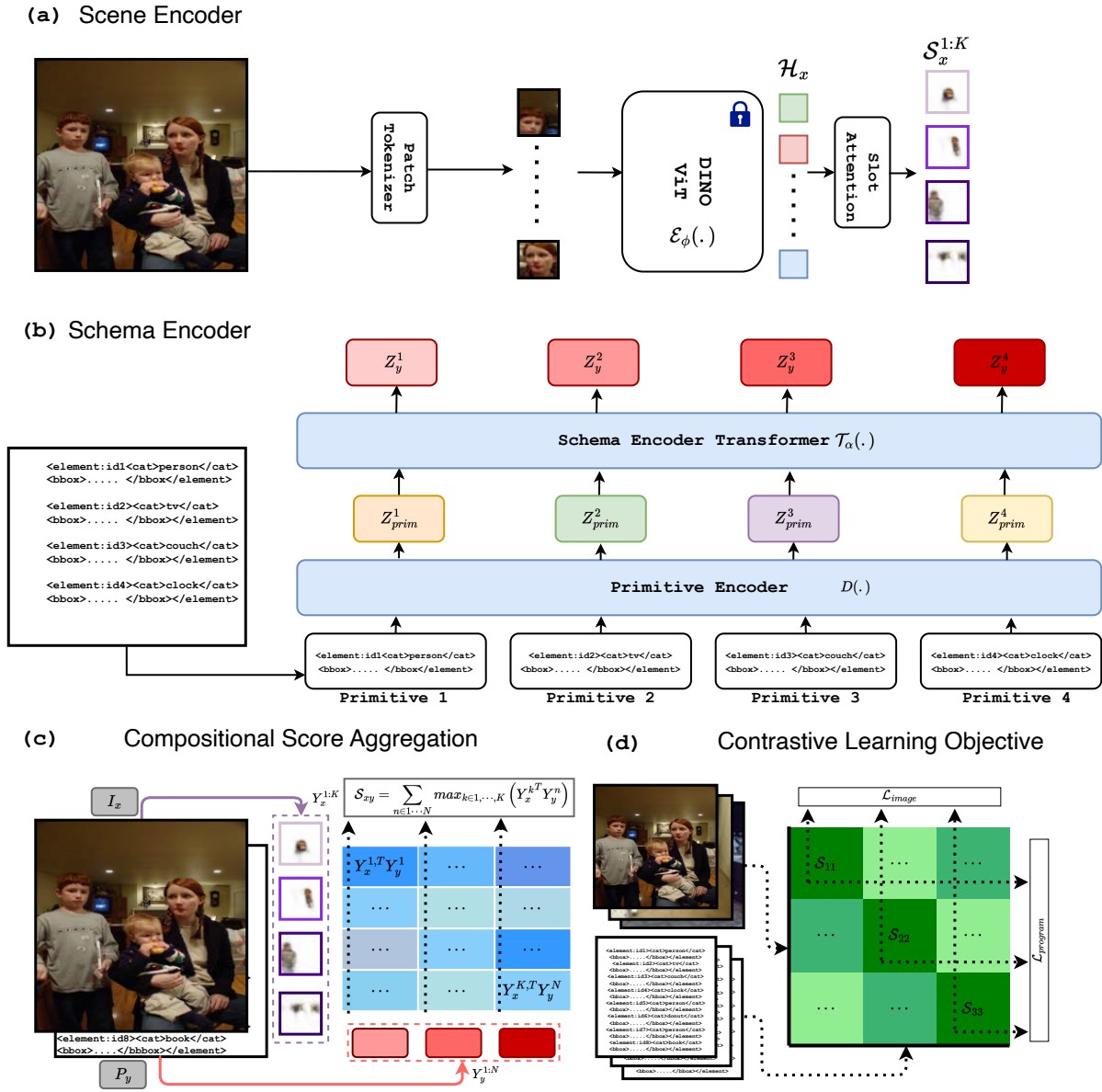

Figure 4: NSI method. (a) A DINOSAUR encoder (Seitzer et al., 2023) learns to represent images via slots. (b) A bi-level schema encoder learns a representation of schema primitives. The primitive encoder embeds the object properties of each schema primitive. Then, a Transformer learns embeddings that assimilate the entire schema context. (c) The inner loop of the metric computes the score $S_{xy}$ between compositional abstractions of an image $I_x$ and a schema $P_y$. Object slots and schema primitives are projected onto a shared embedding space and every latent primitive is assigned to its nearest slot for score aggregation. (d) The $S_{xy}$ scores obtained from *local* entities are used to optimize a *global* contrastive learning objective in the outer loop over a batch of image-schema pairs.

# 5   Experiments

In this section, we set up experiments to (1) answer whether the slots are coherently imbued into the notion of the object properties, (2) study how the grounding compares to traditional weakly supervised contrastive

---

**Algorithm 1** Neural Slot Interpreter Contrastive Learning Pseudocode

---

1: **Require:** Projection heads $\mathcal{H}_{scene}(.), \mathcal{H}_{schema}(.)$
2: **Require:** Batch of slot embeddings from the slot encoder $\{S_x^{1:K}\}_{1:B}$
3: **Require:** Batch of primitive embeddings from the schema encoder $\{Z_y^{1:N}\}_{1:B}$
4: # Compositional Score Aggregation
5: $\{Y_x^{1:K}\}_{1:B} = \mathcal{H}_{scene}\left(\{S_x^{1:K}\}_{1:B}\right)$            ▷ Project slots
6: $\{Y_y^{1:N}\}_{1:B} = \mathcal{H}_{schema}\left(\{Z_y^{1:N}\}_{1:B}\right)$            ▷ Project primitives
7: **for all** $i, j \in \{1, \cdots, B\}$ **do**            ▷ Compute $\mathcal{S} \in \mathbb{R}^{B \times B}$
8:      $\mathcal{S}_{ij} = \sum_{n \in \{1,\cdots,N\}} \max_{k \in \{1,\cdots,K\}} Y_i^{k^T} Y_j^n$
9: **end for**
10: # Contrastive Learning
11: $labels = arange(B)$
12: $\mathcal{L}_{schema} = CrossEntropyLoss\left(labels, \mathcal{S}\right)$
13: $\mathcal{L}_{scene} = CrossEntropyLoss\left(labels, \mathcal{S}^T\right)$
14: $\mathcal{L}_{contrastive} = (\mathcal{L}_{schema} + \mathcal{L}_{scene})/2$
15: **return** $\mathcal{L}_{contrastive}$

---

learning, (3) estimate the quality of alignment under different annotation regimes, (4) probe whether the NSI objective leads to the emergence of slots that bind to raw object features, and (5) answer if the slots are effectively grounded, do they enhance downstream performance?

## 5.1 Schema Instantiation and Architecture Backbone

Our experiments encompass different tasks on scenes ranging from synthetic renderings to in-the-wild scenes viz. (1) CLEVr Hans (Stammer et al., 2020): objects scattered on a plane, (2) CLEVrTex (Karazija et al., 2021): textured objects placed on textured backgrounds (3) MOVi-C (Greff et al., 2022): photorealistic objects on real-world surfaces, and (4) MS-COCO 2017 (Lin et al., 2015): a large-scale object detection dataset containing real-world images. In a pre-processing step, we organize scene labels into the proposed schema (Section 4.1). The property tags `<p_j>` that populate schema primitives in each dataset are listed in Table 1. We use the schema instances to ground object information in their corresponding scenes via NSI. See Appendix A.3 for schema instances.

| Dataset | Property tags `<p_j>` |
|---|---|
| CLEVr Hans | `<color>, <shape>, <material>, <size>, <3D position>` |
| CLEVrTex | `<texture>, <shape>, <size>, <3D position>` |
| MOVi-C | `<category>, <scale>, <2D position>, <bounding box>` |
| MS-COCO 2017 | `<category>, <bounding box>` |

Table 1: Property tags used to instantiate and ground schema primitives.

We followed the DINOSAUR (Seitzer et al., 2023) recipe for learning slot representations. DINOSAUR uses semantically-informative DINO ViT (Caron et al., 2021) features as an autoencoding objective, significantly improving real-world object-centric learning abilities. More specifically, we train an MLP decoder to reconstruct features from slots for all our experiments. Training details and hyperparameters are given in Appendix C.2.

## 5.2 Grounded Compositional Semantics

### 5.2.1 Scene-Schema Alignment Evaluation

Grounded concepts should be effectively aligned to the slots they represent, and we operationalize this ability through the lens of retrieval, a task providing a direct and quantifiable test of this concept-slot alignment. To this end, we set up a bimodal scene-property retrieval task.

**Experiment Setup:** This task consists of two databases - an annotation database and a scene database. The annotation database contains a collection of object properties, and the other contains a collection of scene images. Each set of properties in the annotation database has a corresponding scene in the scene database. In the first half, we measure property retrieval, where models are tasked with retrieving the correct set of object properties, given the scene, using their respective alignment scores. In the second half, the task is inverted to perform scene retrieval, where the models search for scenes in the scene database, given the object properties. We set up the task on CLEVrTex, MOVi-C, and MS-COCO datasets using their test split to instantiate the databases. The task contains 10000, 6000, and 4952 instances in both retrieval databases for CLEVrTex, MOVi-C, and MS-COCO, respectively.

**Baseline Setup:** We consider several baselines for the retrieval task spanning the compositionality continuum.

(a) **CLIP embeddings:** (Radford et al., 2021) CLIP embeddings have been explicitly trained for retrieval on image-text pairs and form a strong non-compositional baseline. Here, we encoded the schema as a string and measured the similarity between the text-image CLIP embeddings.

(b) **GroundingDINO** (Liu et al., 2024): We consider GroundingDINO, which is an open set vision-language model pre-trained on a large corpus to align text and images on a grounding objective. Unlike CLIP, which is non-compositional, the alignment in GroundingDINO is dependent on the compositional sub-alignment between text phrases and image regions. We directly measure the similarity score outputs obtained from the model on a (schema string, image) pair to perform retrieval.

(c) **Ungrounded slots** (Seitzer et al., 2023): Our first slot-based baseline utilizes the slots from a frozen pre-trained DINOSAUR backbone with a shallow predictor fine-tuned for property prediction using HMC (Kuhn, 1955). Subsequently, we used HMC scores for retrieval. See Appendix C.1 for training details.

(d) **HMC matching:** For this baseline, we fine-tune the ungrounded slot architecture, including the backbone end-to-en,d to predict object properties from slots. We used the optimal HMC scores of fine-tuned slots for retrieval.

(e) **Ablations:** On NSI, where **NSI-ResNet 34** replaced DINO ViT with the ResNet backbone, as described in (Elsayed et al., 2022), and on **NSI-Schema Agnostic** where schema primitives were encoded without the schema Transformer (Fig 4 (b)).

**Metrics:** We report Recall@K ($k \in [1, 5]$), which measures the fraction of times a correct item was retrieved among the top $K$ results. The results of the retrieval task are in Fig. 5(a),(b),(c), and Fig. 5(d),(e),(f) visualizes the alignment learned by NSI. We now highlight the major observations.

(a) **Explicitly grounding slots through NSI significantly enhances semantic alignment compared to set matching approaches:** HMC matching slots that rely on set-matching of predicted slots exhibit notably weaker performance across all datasets in both property and scene retrieval tasks. NSI and its ablations, by design, learn to associate individual slots with semantically meaningful object attributes. This explicit grounding mechanism proves crucial, especially as we move towards more complex and realistic datasets like MOVi-C and MS-COCO, where the performance gap between NSI-based models and set-matching methods widens considerably.

(b) **The full model is essential:** Schema-agnostic encoders that encode object primitives independently perform sub-par compared to NSI. This underscores the importance of contextualizing object semantics in the overall schema. The pre-trained DINO backbone is crucial for textured objects and real-world generalization, as evidenced by CLEVrTex/COCO results.

(c) **Vision-Language Models grounded in text inadequately capture object semantics:** CLIP embeddings and GroundingDINO both fall short in retrieving object properties beyond basic semantic categories, as evidenced by the weaker performance on MOVi-C and CLEVrTex datasets. This limitation

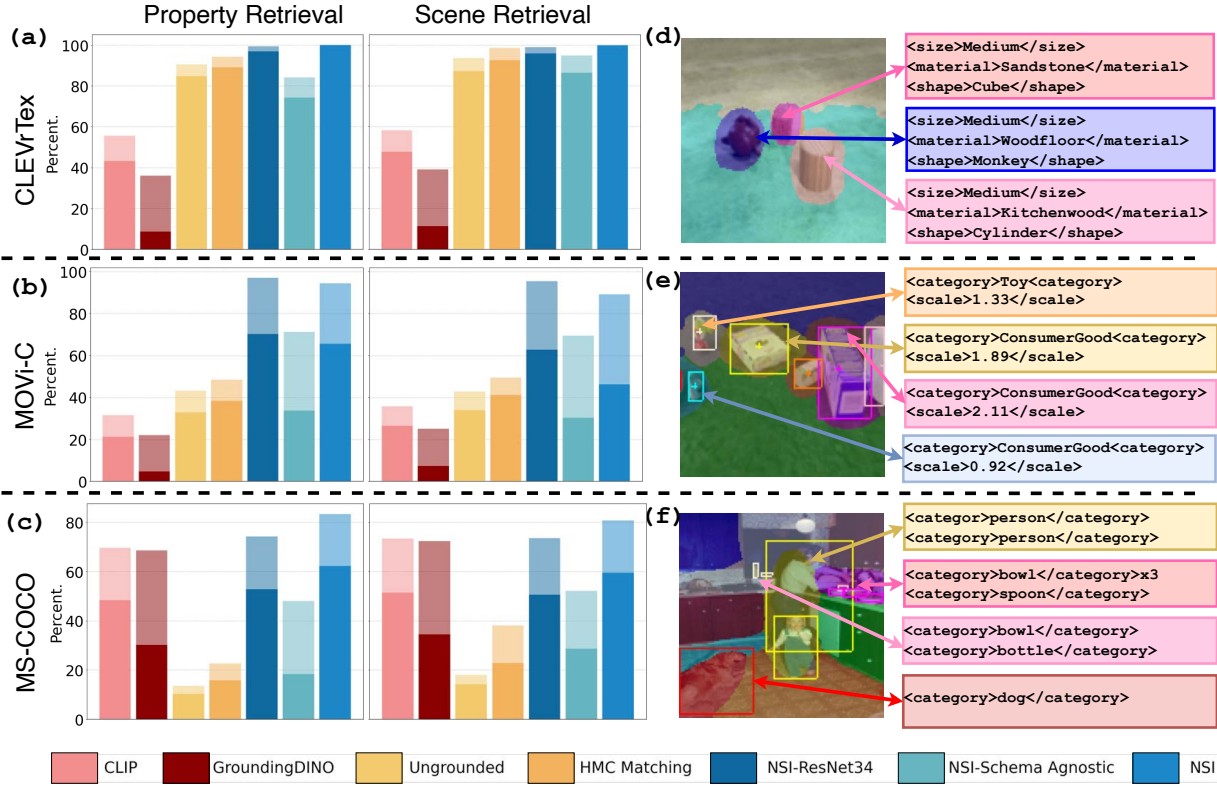

Figure 5: Retrieval results. (a), (b), (c) Property and scene retrieval results. We report Recall@1/5 (higher is better). The standard deviation (over five seeds) was < 0.3 across all model instances and retrieval tasks. The lighter shade shows Recall@5 while the darker shade shows Recall@1. (d), (e), (f) Visualization of correspondences learned by the NSI similarity metric. The colored arrows show the respective correspondences of schema primitives to the slots. Each schema instance is chunked and color-coded by the slot to which its primitives are assigned.

is further emphasized by their relatively weak performance across the two datasets compared to methods that explicitly model object compositionality in scene and annotation representations. We posit that while natural language is effective in grounding object properties pertaining to broad categories (as evidenced by the stronger performance on MS-COCO), it appears insufficiently equipped for object-centric grounding (Chandu et al., 2021).

**Qualitative:** In Fig. 5(d),(e),(f), we visualize the slot-object pairs inferred by our scoring metric. Interpretable and dense correspondences emerge from NSI contrastive learning. Notably, in complex real-world COCO scenes (Fig. 5(f)), NSI slots successfully ground to a diverse set of objects, even capturing instances where a single slot might be associated with multiple related objects (e.g., multiple bowls and a spoon associated with a slot). See Appendix D.1.6 for more results.

### 5.2.2 Comparison against Traditional Weakly-Supervised Visual Grounding

Weakly supervised visual grounding, by learning to link phrases in a caption to specific regions in an image, is a well-established area of research (Karpathy & Fei-Fei, 2015; Yeh et al., 2018; Wang & Specia, 2019; Gupta et al., 2020). These same region-phrase alignment techniques can be easily adapted to the scene-schema grounding problem by replacing phrase embeddings with schema representations. However, a key distinction between these approaches and NSI is how scenes are represented. Traditional image-caption alignment methods rely on pre-trained models to propose bounding boxes around objects, while NSI learns slot representations concurrently with the grounding objective. In this section, we then explore (1) whether

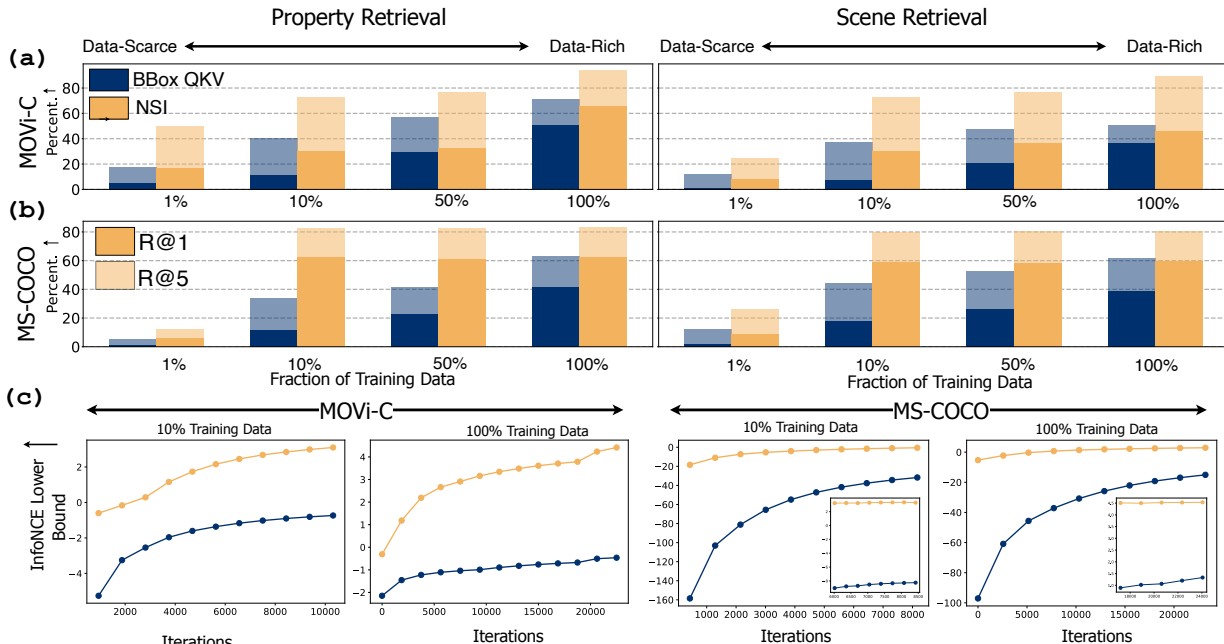

Figure 6: Comparison against traditional weakly supervised visual grounding. We compared NSI against a state-of-the-art bounding box-based phrase grounding method (Gupta et al., 2020). (a) and (b) compare the performance of either method on the property-scene retrieval task on MOVi-C and COCO datasets, respectively. Despite being trained on the same InfoNCE objective, NSI outperforms region-based grounding, demonstrating that slot representations emergent from the NSI objective are key to grounding object semantics. Moreover, NSI is also data-efficient in real-world COCO scenes. (c) plots the InfoNCE lower bound attained on the test set as training progresses (smoothened for improved visualization). NSI has a stronger lower bound and more effectively captures the scene-schema MI via object-centric abstractions.

bounding box representations are equally effective as slot representations at grounding object properties, and (2) how the efficacy of each method compares across different annotation levels.

**Experiment Setup:** We investigated the efficacy of the contrastive learning-based phrase grounding method by Gupta et al. (2020) and trained it on scene-schema pairs from the MOVi-C and COCO datasets. This method utilizes bounding box proposals to encode scenes, and we refer to it as **BBox QKV**. Like NSI, BBox QKV optimizes the modality encoders to maximize the lower bound on MI via the InfoNCE loss. In addition, this method trains extra Query, Key, and Value heads to compute similarity scores between image regions and schema primitives via the attention method.

**Baseline Setup:** Bounding box-based grounding methods rely on detectors like Fast R-CNN (Girshick, 2015) to extract image regions. To simplify the setup and eliminate errors arising from the detection pipeline, we directly utilize ground truth bounding box annotations provided in the schema. The bounding boxes are then represented by directly averaging the DINO features contained within the box and used to ground scenes into schema labels.

**Dataset Setup:** We further sought to understand the data efficiency of NSI against the conventional bounding box-based approach. To this end, we controlled for training data size and explored four training regimes for each method: {1%, 10%, 50%, and 100% } of the training dataset, ranging from data-scarce to data-rich.

**Evaluation Setup:** We evaluated BBox QKV trained on various dataset sizes on the bimodal retrieval task setup from the previous section. The retrieval results are summarized in Fig. 6. Our major findings were:

(a) **Slot encoding outperforms Bounding Box encoding.** NSI consistently outperforms BBox QKV in both Property and Scene Retrieval tasks on either dataset (see Fig. 6(a) and (b)). This performance

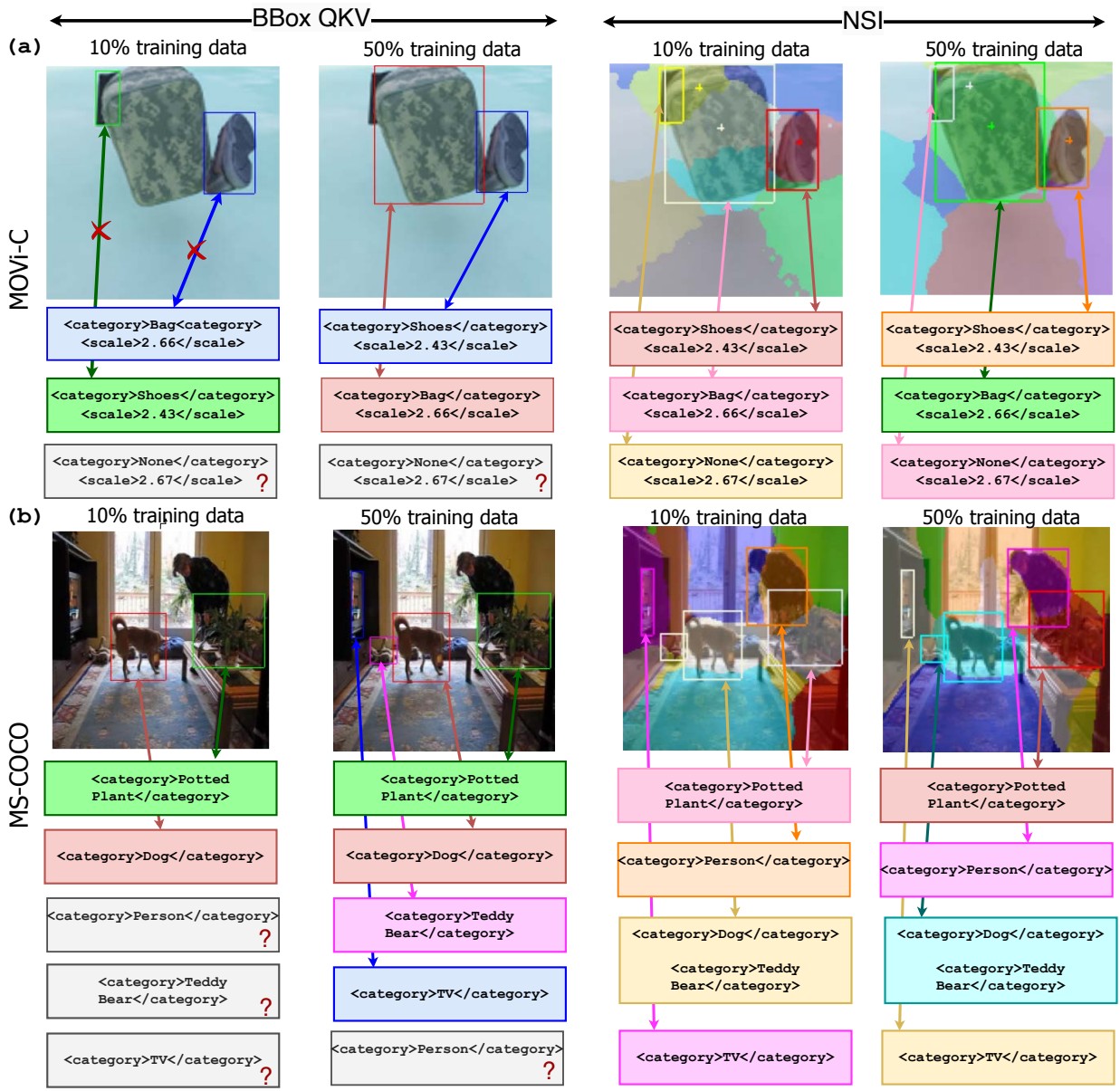

Figure 7: BBox QKV vs NSI retrieval results. After being trained on two different settings, we demonstrate the emergent correspondences from BBox QKV and NSI. For NSI, at the 10% setting, the slots are less robust than the 50% setting, however, the correspondences between objects in the slot and their labels are accurate overall. On the other hand, we observe multiple failure modes for BBox-QKV at low-data settings where overlapping regions are assigned incorrectly or the model fails to ground objects at all (see (a) and (b), BBox QKV 10% training data). The issue with disambiguating occluded and overlapping objects persists even with greater training data.

difference arises despite both methods employing identical InfoNCE estimation and scoring functions, differing solely in image encoding: NSI utilizes slots, while BBox QKV relies on bounding boxes. This suggests that coarser bounding boxes may underspecify objects they contain, leading to weaker semantic grounding and diminished retrieval efficacy than slot-based counterparts. On the other hand, the slot-attention mechanism explicitly encourages symmetry breaking between slots such that they encode non-overlapping fine-grained spatial features that facilitate improved alignment to object features.

(b) **NSI is data-efficient on real-world scenes.** NSI exhibits remarkable data efficiency, demonstrating robust performance even in data-scarce scenarios, notably on MS-COCO. Performance remains strong with only 10% ($\sim$ 10,000 examples) of MS-COCO training data. Despite limited data, this robustness highlights that grounding via the NSI objective facilitates a strong transfer of discriminative object semantics to slots that originally only contained spatial biases. Moreover, such transfer is significantly more data-efficient compared to the performance of bounding box abstractions across dataset sizes.

(c) **NSI achieves a higher InfoNCE lower bound.** We plot the InfoNCE lower bound (Eq. 17) obtained from the validation split across the training iterations in Fig. 6(c). NSI consistently achieves a much stronger (higher) lower bound compared to BBox QKV for both MOVI-C and MS-COCO datasets and across annotation levels. A higher InfoNCE Lower Bound for NSI indicates that it is more effectively extracting the MI between schema and scene pairs. This further suggests that the slot-based representations learned by NSI allow for a more discriminative representation of the relationship between scenes and their object semantics.

**Qualitative:** Fig. 7 (a) and (b) demonstrate the correspondences learned by NSI and BBox QKV across the datasets and the 10% and 50% training data fractions. The qualitative results further illuminate the limitations of bounding box representations, particularly in scenarios with object overlap or occlusion. As seen in MOVI-C (a) and MS-COCO (b), BBox QKV exhibits failure modes when objects are not distinctly isolated, often misattributing objects (e.g., see MOVI-C 10% data) or failing to ground objects altogether. These instances suggest that bounding boxes, being granular and encompassing potentially overlapping regions, struggle to disambiguate objects and their semantics, especially when trained on fewer instances.

### 5.3 Object Discovery with NSI

Object-centric frameworks have been traditionally used to bind neural network representations to distinct objects within a scene. Here, we evaluate whether grounded slots are more adept at discovering objects in the context of visual segmentation.

**Experiment Setup:** We extract object attention masks from the backbone derived from slot-attention clusters. The masks are then upsampled from the feature resolution to image resolution. We compared these masks to the ground truth segmentation across the CLEVrTex, MOVi-C, and MS-COCO datasets on the test split scenes.

**Baselines Setup**: We compare masks from the DINOSAUR scene encoder trained on three different grounding objectives.

(a) **Ungrounded slots** derived from the autoencoder reconstruction objective, without any semantic grounding.
(b) **HMC matching** grounded slots learned from predicting object properties from slots via set-matching.
(c) **NSI** grounded slots that explicitly ground object properties in slots via latent semantic assignment.

**Metrics:** We report the quality of object masks obtained from the slot representations based on two segmentation metrics: (1) Foreground Adjusted Rand Index (**FG-ARI**): measures the accuracy of clustering foreground objects into their respective segments and (2) Mean Best Overlap (**mBO**): assesses the best overlap between predicted and ground truth object masks. We report both instance ($mBO^i$) and class ($mBO^c$) level $mBO$ scores for COCO. The segmentation results can be found in Fig. 8(a). We observe that:

(a) **NSI endowed slots meaningfully segment objects.** Object masks generated by NSI are competitive on synthetic scenes and markedly improve object discovery on COCO scenes. We posit that contrastive learning via NSI enhances symmetry breaking of the slot attention backbone for challenging real-world scenes, effectively binding slots to raw visual features.

(b) **HMC matching obscures object discovery on COCO.** Constraining slots to predict a single object forces the backbone to develop specialized representations for each object. This imposes a difficult learning problem on real-world scenes and causes slots to deteriorate.

(c) **NSI is biased towards semantic classes over instance classes.** On COCO instances, semantic segmentation scores ($mBO^c$) are higher compared to the baseline. We attribute this to the NSI metric that biases slots to represent broader categories by grounding multiple objects.

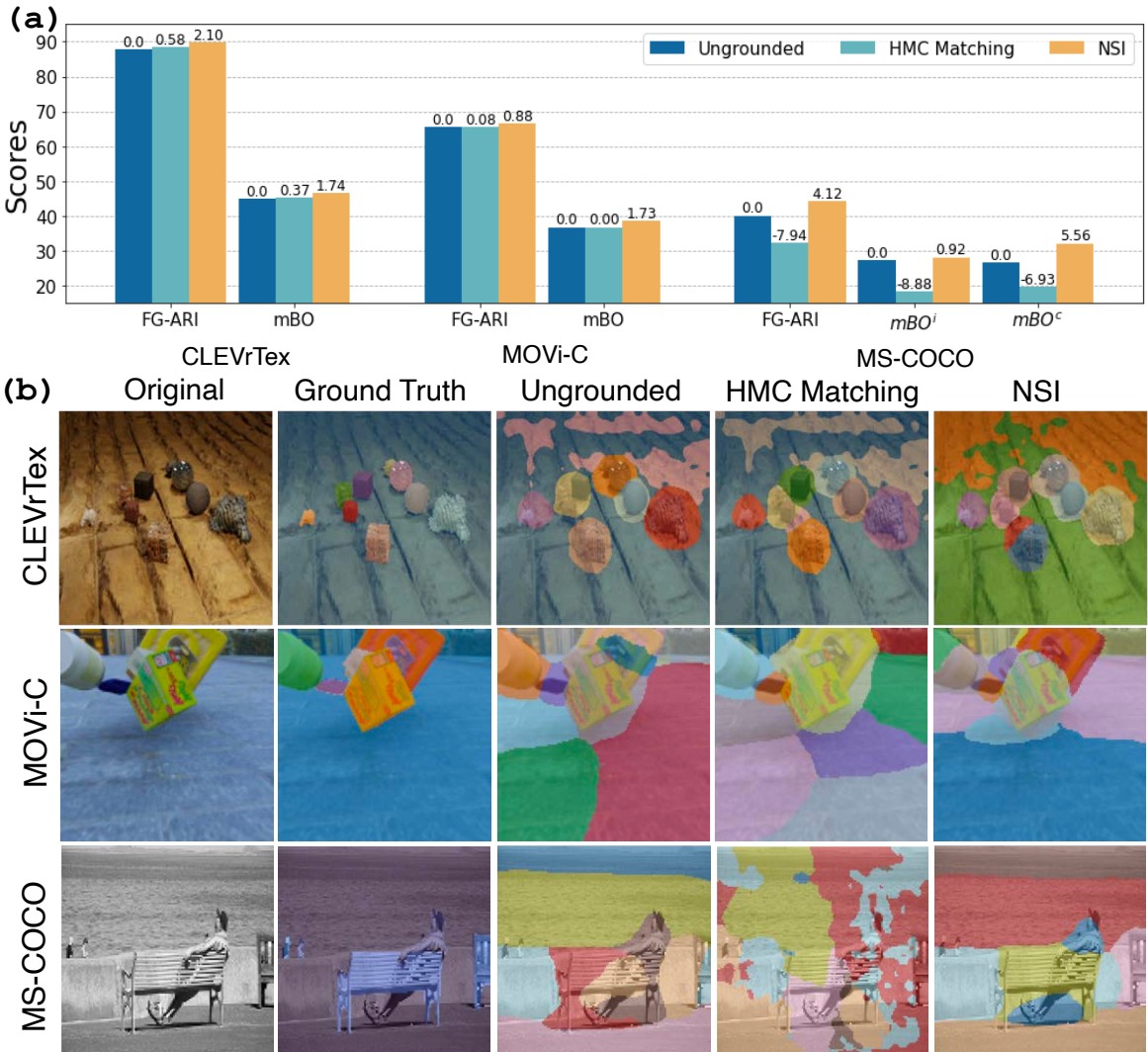

Figure 8: Object discovery results. (a) Segmentation results on three datasets. We report scores of *FG-ARI* (higher is better) and *mBO* (higher is better). We also show scores relative to the ungrounded baseline on top of each bar. (b) Visualization of attention masks learned by different models on instances of the datasets. See Appendix D.2.1 for error bars and more qualitative results.

**Qualitative:** Fig. 8(b) visualizes the object masks that are emergent across the various methods. We notice that NSI grounding of object semantics is helpful in disambiguating masks that violate object boundaries or contain more than one object. For example, in the CLEVrTex example, the yellow mask learned by the Ungrounded instance contains two objects, but the NSI instance, which encourages each slot to map to its distinct semantics, separates the two objects. Similarly, in the case of the COCO example, the boundaries of the bench and person are obfuscated and ambiguously overlapped. However, in the NSI instance, the learned mask of the person is non-contiguous and allows for the full bench mask to form.

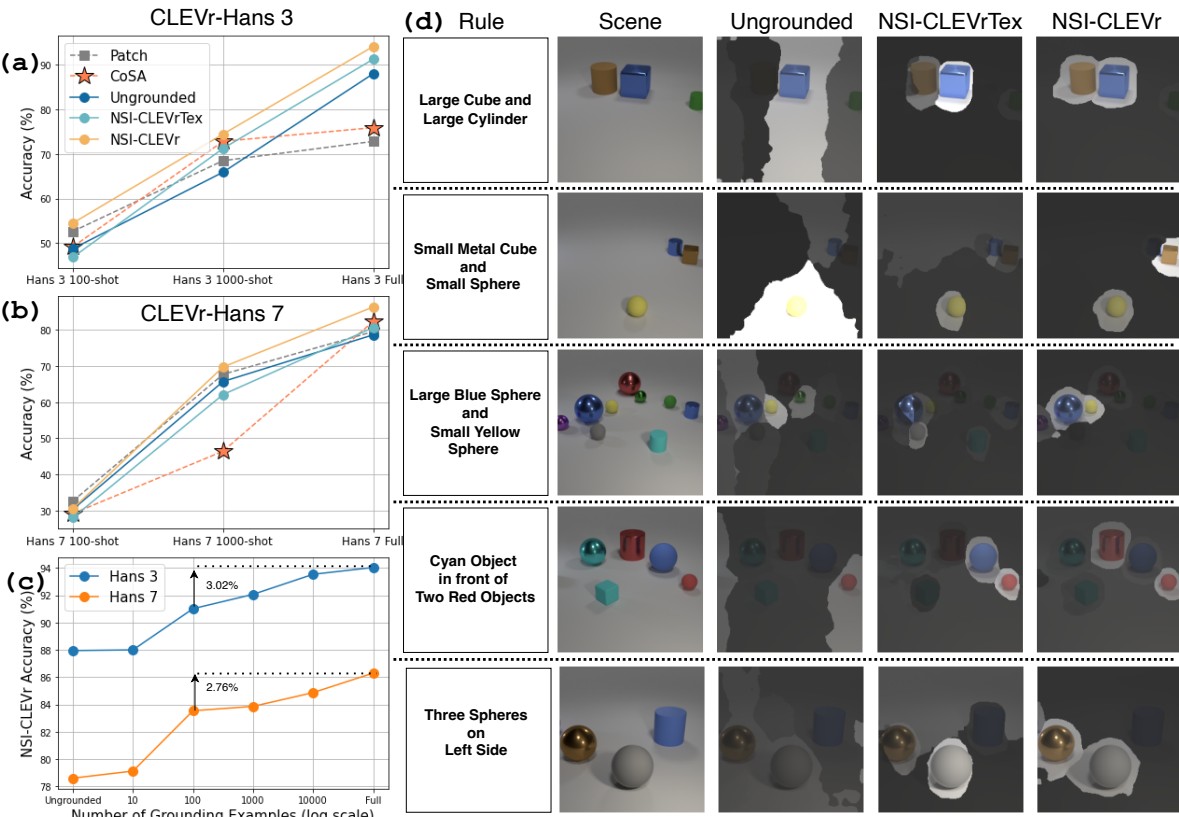

Figure 9: Few-shot classification results. (a) (b) Few-shot classification accuracy on Hans 3 and Hans 7 datasets, respectively. (c) Task accuracy on NSI-CLEVr tokenizers trained on different numbers of grounding examples. The standard deviation (over five seeds) was $< 0.03$ across tasks and methods. (d) Visual rationales are generated across the grounding continuum by extracting attention maps from the final ViT layer. See Appendix D.3.2 for more examples.

### 5.4 Grounded Slots as Visual Tokens

Grounding-agnostic patch-based tokens are the *de facto* standard for transformer-based models. On the other hand, humans can flexibly abstract out entities free of rigid geometric templates. Here, we investigate the ability of grounded slots to bridge this abstraction gap on a downstream classification task.

**Experiment Setup:** We consider the CLEVr-Hans classification benchmark (Stammer et al., 2020). The data set comprises images derived from the CLEVr scenes, which are partitioned into distinct classes. Class membership is determined by predefined combinations of object attributes and their inter-object relationships (e.g., 'Large cube and large cylinder'). Notably, specific classes within this collection are intentionally designed with confounding factors that consists of multiple classes based on object attributes and relations. Moreover, the true membership properties are confounded with other attributes in the train split. Consequently, achieving accurate classification on test images necessitates that the model effectively deconfound the attribute correlations inherent in the training set. Here, we maintain a common ViT classifier (Dosovitskiy et al., 2021) backbone and train it on tokens obtained from different tokenizers. We conduct experiments on CLEVr-Hans 3 and CLEVr-Hans 7 benchmarks containing three and seven classification categories, respectively. We further sweep across training dataset size across $100, 1000$, and the full training set.

**Baselines Setup:** We explore the following tokenization schemes:

(a) Traditional **patch** tokens extracted from $14 \times 14$ image patches of the scene.

(b) **Ungrounded** slot tokens learned from autoencoder object-centric learning on CLEVr.

(c) **Conditional Slot Attention (CoSA)** slot tokens (Kori et al., 2024a) derived from set-matching slots to property labels on the CLEVr dataset.

(d) **NSI-CLEVrTex** slot attention trained on CLEVrTex via NSI. The backbone is frozen and subsequently used to infer CLEVr slots. This setup makes the slots *partially grounded* because CLEVr and CLEVrTex share common attribute types (shape, size, texture), and the scenes and the scenes across the datasets are also structurally similar.

(e) *Fully grounded* **NSI-CLEVr** slot attention trained on CLEVr via NSI.

**Metrics:** In Fig. 9(a),(b), we report test accuracy against the $k$-shot training sweep for CLEVr-Hans 3 and CLEVr-Hans 7. We find that:

(a) **Grounded slots facilitate improved reasoning** and surpass patch-based tokenizers across data regimes using $25\times$ fewer tokens. While the performance of the patch-tokens saturates, grounded slots de-confound object attributes with greater ability with increasing data.

(b) **Partial grounding is often sufficient:** NSI-CLEVrTex slots grounded in a different dataset demonstrate competitive reasoning abilities. CLEVrTex and CLEVr capture the common structural notion of objects in a scene but significantly differ in their texture semantics. However, a tokenizer trained on CLEVrTex still facilitates downstream reasoning on the CLEVr scenes. This result further corroborates the strong transfer effect of explicit grounding observed in Section 5.2.2 where grounded slots assimilate generalizable representations with annotations from the same scene and also facilitate rapid transfer across distinct stimuli.

(c) **Property prediction ability does not necessarily transfer to class prediction:** CoSA slots trained on object-attribute prediction learned from HMC matching show weaker transfer on the Hans-3 dataset. This points out another limitation of the one-to-one template imposed by HMC where slots are predictive of individual objects but lack the contextual abstractions for downstream inference.

**How many schema annotations are essential?** In Fig. 9(c), when ablating the number of annotations used to train the NSI-CLEVr tokenizer, we observe that inductive biases instilled by grounding are key, as seen from the sensitivity of the performance to the number of examples. On the other hand, significant annotation of scenes is not necessary. Annotating just 100 schema examples yields performative accuracy within a 3% margin of the tokenizer trained on the complete set.

**Qualitative:** We probed the attention maps from the final layer of the ViT to generate visual rationales. Fig. 9(d) visualizes the maps across the grounding continuum. The ungrounded slots showed little to no correlation with the class rule. The partially ground slots in CLEVrTex are more discriminative than the ungrounded slots but often yield incorrect rationales. On the other hand, ViT trained on the NSI-CLEVr slots and weighed slots pertinent to the class rule with greater attention. The visual rationales can be valuable in interpreting the predictions of the model.

## 6 Conclusion

This work introduced NSI, which grounds object semantics into slots for object-centric understanding. It uses a simple schema abstraction to define object concepts and learns to flexibly associate neural embeddings of the schema primitives with object slots via contrastive learning. NSI facilitates interpretable grounding in slot representations. Whereas natural language-grounded embeddings struggle to retrieve granular object properties, NSI embeddings abstracted from the slot-schema intermodal alignment are key representations for such tasks. The effectiveness of the representations is also highlighted when pre-specified bounding boxes trained with the same contrastive learning objective demonstrate weaker grounding. Furthermore, unlike set-matching approaches, which struggle when scaled to real-world scenes, NSI enhances object discovery compared to ungrounded counterparts. Finally, we demonstrated the usefulness of NSI as a grounding-aware visual tokenizer that improves the few-shot visual reasoning abilities of ViTs on a hard classification task. The major limitation of the work lies in requiring annotations that can be prohibitively expensive and laborious. However, we empirically showed the data efficiency of the method on real-world scenes where using as little as 10% training data yielded grounding comparable to the full setting. We also demonstrated performative reasoners under practical annotation settings. Slot representations have traditionally been associated with

visual stimuli, but object concepts transcend perception to other sensorimotor experiences like audio, tactile signals, and motor behaviors. To this end, NSI lays the groundwork for multimodal object-centric learning. Future work involves adopting NSI to ground common object-centric concepts into different sensorimotor experiences as a step towards a modular human-like understanding of the world.

## 7 Acknowledgment

This work was supported by NSF under grant no. CCF2203399. The experiments reported in this article were performed on the computational resources managed and supported by Princeton Research Computing at Princeton University. We also express gratitude to The Sacred Grounds Cafe and Karma Cafe in San Francisco for brewing the best coffee North of the Panhandle.

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

# Appendix

## A   Data

This section contains additional information on the datasets, nested schema space instantiation, and examples of the annotation organization.

### A.1   Nested Schema Space Description

Table 2 lists the various object properties used to create schema description of scenes.

### A.2   Dataset Splits

The dataset splits used in this work are detailed in Table 3.

### A.3   Schema Examples

Figs. 10-13 show schema descriptions of instances from all the datasets.

| Dataset | Property | Discrete/Continuous | Size |
|---|---|---|---|
| CLEVr Hans (Johnson et al., 2016; Stammer et al., 2020) | Material | Discrete | 2 |
|  | Color | Discrete | 8 |
|  | Shape | Discrete | 3 |
|  | Size | Discrete | 2 |
|  | Object Position | Continuous | 3 |
| CLEVrTex (Karazija et al., 2021) | Texture | Discrete | 60 |
|  | Shape | Discrete | 4 |
|  | Size | Discrete | 3 |
|  | Object Position | Continuous | 3 |
| MOVi-C (Greff et al., 2022) | Object Category | Discrete | 17 |
|  | Object Size | Continuous | 1 |
|  | Object Position | Continuous | 2 |
|  | Bounding Box | Continuous | 4 |
| MS-COCO 2017 (Lin et al., 2015) | Object Category | Discrete | 90 |
|  | Bounding Box | Continuous | 4 |

Table 2: Schema space across various datasets.

| Name | Train Split Size | Validation Split Size | Test Split Size |
|---|---|---|---|
| CLEVr Hans 3 | 9000 | 2250 | 2250 |
| CLEVr Hans 7 | 21000 | 5250 | 5250 |
| CLEVrTex | 37500 | 2500 | 10000 |
| MOVi-C | 198635 | 35053 | 6000 |
| MS COCO 2017 | 99676 | 17590 | 4952 |

Table 3: Dataset splits used in experiments.

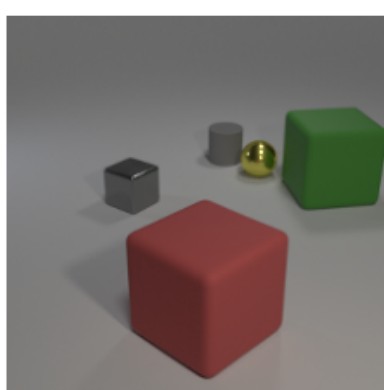

```
<element: 0>
    <size> small</size>
    <shape> cube</shape>
    <color> gray</color>
    <material> metal</material>
    <pos> (-0.76, -0.79, 0.35)</pos>
</element>
<element: 1>
    <size> small</size>
    <shape> sphere</shape>
    <color> yellow</color>
    <material> metal</material>
    <pos> (-0.14, 1.75, 0.35)</pos>
</element>
<element: 2>
    <size> large</size>
    <shape> cube</shape>
    <color> green</color>
    <material> rubber</material>
    <pos> (1.24, 2.12, 0.69)</pos>
</element>
<element: 3>
    <size> small</size>
    <shape> cylinder</shape>
    <color> gray</color>
    <material> rubber</material>
    <pos> (-1.11, 1.79, 0.35)</pos>
</element>
<element: 4>
    <size> large</size>
    <shape> cube</shape>
    <color> red</color>
    <material> rubber</material>
    <pos> (2.73, -2.23, 0.69)</pos>
</element>
```

Figure 10: CLEVr Hans instance with its corresponding schema description.

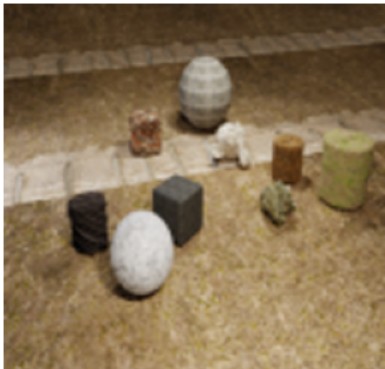

```
<element:0><size>medium</size>
    <shape> sphere</shape>
    <material> whitemarble</material>
    <pos> (1.58, -2.81, 0.60) </pos>
</element>
<element:1><size>small</size>
    <shape> cylinder</shape>
    <material> polyhaven_aerial_mud_1</material>
    <pos> (-0.23, -2.94, 0.40) </pos>
</element>
<element:2><size>medium</size>
    <shape> cylinder</shape>
    <material> polyhaven_forrest_ground_01</material>
    <pos> (2.67, 2.78, 0.60) </pos>
</element>
<element:3><size>medium</size>
    <shape> monkey</shape>
    <material> polyhaven_cracked_concrete_wall</material>
    <pos> (-0.63, 1.98, 0.60) </pos>
</element>
<element:4><size>medium</size>
    <shape> cube</shape>
    <material> polyhaven_brick_wall_005</material>
    <pos> (-2.81, 0.52, 0.42) </pos>
</element>
<element:5><size>large</size>
    <shape> sphere</shape>
    <material> polyhaven_large_grey_tiles</material>
    <pos> (-2.66, 2.94, 0.90) </pos>
</element>
<element:6><size>small</size>
    <shape> cylinder</shape>
    <material> polyhaven_leaves_forest_ground</material>
    <pos> (1.12, 2.49, 0.40) </pos>
</element>
<element:7><size>small</size>
    <shape> monkey</shape>
    <material> polyhaven_aerial_rocks_01</material>
    <pos> (1.98, 0.84, 0.40) </pos>
</element>
<element:8><size>medium</size>
    <shape> cube</shape>
    <material> polyhaven_wood_planks_grey</material>
    <pos> (0.78, -1.23, 0.42) </pos>
</element>
```

Figure 11: CLEVrTex instance with its corresponding schema description.

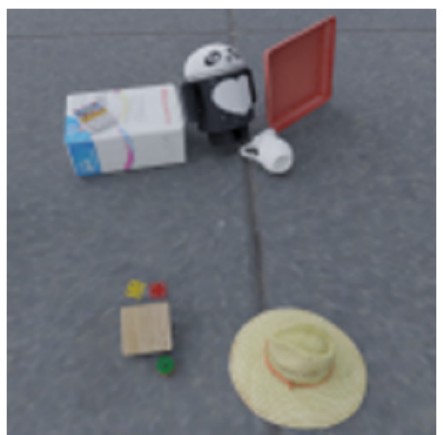

```
<element:0><category>Hat</category>
    <scale> 1.93 </scale>
    <position> (0.68, 0.80)  </position>
    <bbox> (0.70, 0.52, 0.98, 0.84) </bbox>
</element>
<element:1><category>Consumer Goods</category>
    <scale> 2.25 </scale>
    <position> (0.28, 0.28)  </position>
    <bbox> (0.18, 0.14, 0.39, 0.42) </bbox>
</element>
<element:2><category>None</category>
    <scale> 1.98 </scale>
    <position> (0.67, 0.15)  </position>
    <bbox> (0.02, 0.60, 0.29, 0.77) </bbox>
</element>
<element:3><category>Consumer Goods</category>
    <scale> 1.97 </scale>
    <position> (0.50, 0.20)  </position>
    <bbox> (0.09, 0.41, 0.32, 0.58) </bbox>
</element>
<element:4><category>Toys</category>
    <scale> 1.45 </scale>
    <position> (0.33, 0.75)  </position>
    <bbox> (0.63, 0.27, 0.86, 0.39) </bbox>
</element>
<element:5><category>Media Cases</category>
    <scale> 0.80 </scale>
    <position> (0.20, 0.25)  </position>
    <bbox> (0.21, 0.15, 0.29, 0.26) </bbox>
</element>
<element:6><category>None</category>
    <scale> 1.02 </scale>
    <position> (0.62, 0.33)  </position>
    <bbox> (0.28, 0.54, 0.38, 0.66) </bbox>
</element>
```

Figure 12: MOVi-C instance with its corresponding schema description.

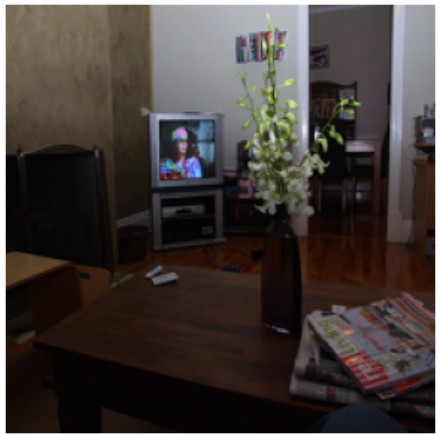

```
<element:0>
    <cat>tv</cat>
    <bbox> (0.34, 0.26, 0.18, 0.17)  </bbox>
</element>
<element:1>
    <cat>chair</cat>
    <bbox> (0.03, 0.32, 0.27, 0.33)  </bbox>
</element>
<element:2>
    <cat>book</cat>
    <bbox> (0.70, 0.67, 0.30, 0.24)  </bbox>
</element>
<element:3>
    <cat>vase</cat>
    <bbox> (0.59, 0.46, 0.10, 0.32)  </bbox>
</element>
<element:4>
    <cat>chair</cat>
    <bbox> (0.72, 0.27, 0.08, 0.22)  </bbox>
</element>
<element:5>
    <cat>dining table</cat>
    <bbox> (0.79, 0.32, 0.07, 0.22)  </bbox>
</element>
<element:6>
    <cat>remote</cat>
    <bbox> (0.34, 0.62, 0.06, 0.04)  </bbox>
</element>
<element:7>
    <cat>book</cat>
    <bbox> (0.71, 0.77, 0.07, 0.09)  </bbox>
</element>
<element:8>
    <cat>chair</cat>
    <bbox> (0.80, 0.28, 0.09, 0.21)  </bbox>
</element>
```

Figure 13: COCO instance with its corresponding schema description.

## B    Neural Slot Interpreter

Next, we present details of NSI training modules and summarize the NSI pseudocode.

## C    Hyperparameters

We list the hyperparameters for NSI and other methods used in our experiments, which were all performed on Nvidia A100 GPUs.

### C.1    Ungrounded and HMC Matching Backbone Hyperparameters

The hyperparameters for ungrounded and HMC matching backbones are given in Table 4.

| Module | Hyperparameters | CLEVr Hans | CLEVrTex | MOVi-C | MS COCO 2017 |
|---|---|---|---|---|---|
| DINO Backbone | Image Size | 224 | 224 | 224 | 224 |
| | Patch Size | 8 | 8 | 8 | 8 |
| | Num. Patches | 784 | 784 | 784 | 784 |
| | Num. Layers | 8 | 8 | 8 | 8 |
| | Num. Heads | 8 | 8 | 8 | 8 |
| | Hidden Dims. | 192 | 192 | 192 | 192 |
| Slot Attention | Num. Slots | 10 | 10 | 10 | 93 |
| | Iterations | 3 | 3 | 3 | 3 |
| | Hidden Dims. | 192 | 192 | 192 | 192 |
| Broadcast Decoder | Num. MLP Layers | 3 | 3 | 3 | 3 |
| | MLP Hidden Dims. | 1024 | 1024 | 1024 | 1024 |
| | Output Dims. | 785 | 785 | 785 | 785 |
| Prediction Head | MLP Hidden Layers | 2 | 2 | 2 | 2 |
| | MLP Hidden Dims. | 64 | 64 | 64 | 64 |
| | Output Size | 18 | 71 | 25 | 95 |
| Training Setup | Batch Size | 128 | 128 | 128 | 128 |
| | LR Warmup steps | 10000 | 10000 | 10000 | 30000 |
| | Peak LR | $4 \times 10^{-4}$ | $4 \times 10^{-4}$ | $4 \times 10^{-4}$ | $1 \times 10^{-4}$ |
| | Dropout | 0.1 | 0.1 | 0.1 | 0.1 |
| | Gradient Clipping | 1.0 | 1.0 | 1.0 | 1.0 |
| Inference Configuration | Num. Slots | 10 | 10 | 10 | 30 |
| Training Cost | GPU Usage | 40 GB | 40 GB | 40 GB | 40 GB |
| | Days | 1 | 3 | 3 | 5 |
| ViT | Num Layers | 2 | - | - | - |
| | Hidden Dims | 64 | - | - | - |
| | Num Heads | 4 | - | - | - |

Table 4: Hyperparameters for the ungrounded and HMC matching method used in our experiments. In the ungrounded case, the backbone and slot attention modules are trained solely on the reconstruction objective and frozen.

### C.2    NSI Hyperparameters

The hyperparameters for the NSI alignment model are listed in Table 5. The ablated architectures follow the same setup without the ablated module.

| Module | Hyperparameters | CLEVr Hans | CLEVrTex | MOVi-C | MS COCO 2017 |
|---|---|---|---|---|---|
| DINO Backbone | Image Size | 224 | 224 | 224 | 224 |
| | Patch Size | 8 | 8 | 8 | 8 |
| | Num. Patches | 784 | 784 | 784 | 784 |
| | Num. Layers | 8 | 8 | 8 | 8 |
| | Num. Heads | 8 | 8 | 8 | 8 |
| | Hidden Dims. | 192 | 192 | 192 | 192 |
| Broadcast Decoder | Num. MLP Layers | 3 | 3 | 3 | 3 |
| | MLP Hidden Dims. | 1024 | 1024 | 1024 | 1024 |
| | Output Dims. | 785 | 785 | 785 | 785 |
| Slot Attention | Num. Slots | 10 | 10 | 10 | 15 |
| | Iterations | 3 | 3 | 3 | 3 |
| | Hidden Dims. | 192 | 192 | 192 | 192 |
| Schema Encoder | Num. Layers | 8 | 8 | 8 | 8 |
| | Num. Heads | 8 | 8 | 8 | 8 |
| | Hidden Dims. | 192 | 192 | 192 | 192 |
| | Max. Schema Len. | 10 | 10 | 10 | 93 |
| Projection Heads | Embedding Dims. | 64 | 64 | 64 | 64 |
| | MLP Hidden Layers | 2 | 2 | 2 | 2 |
| | MLP Hidden Dims. | 256 | 256 | 256 | 256 |
| Training Setup | Batch Size | 128 | 128 | 128 | 128 |
| | LR Warmup steps | 10000 | 10000 | 10000 | 30000 |
| | Peak LR | $4 \times 10^{-4}$ | $4 \times 10^{-4}$ | $4 \times 10^{-4}$ | $1 \times 10^{-4}$ |
| | Dropout | 0.1 | 0.1 | 0.1 | 0.1 |
| | Gradient Clipping | 1.0 | 1.0 | 1.0 | 1.0 |
| | $\beta_1, \beta_2$ | 0.5, 0.5 | 0.5, 0.5 | 0.5, 0.5 | 0.5, 0.5 |
| Training Cost | GPU Usage | 40GB | 40 GB | 40 GB | 40 GB |
| | Days | 1 | 3 | 3 | 5 |
| ViT | Num Layers | 2 | - | - | - |
| | Hidden Dims | 64 | - | - | - |
| | Num Heads | 4 | - | - | - |

Table 5: Hyperparameters for the NSI alignment model instantiation and training setup.

# D   Experiments

## D.1   Grounded Compositional Semantics

### D.1.1   Top Learned Slots

Figs. 14-16 show the top 50 slots learned by the image encoder over each dataset. Each slot is weighted by the $l_2$ norm of its embeddings $Y_x^k$. The top slots follow an interesting distribution. On CLEVrTex (Fig. 14), slots with larger objects and a clean segmentation are assigned a higher magnitude. Intuitively, these slots are the most discriminative when assigning primitives that contain shape, texture, and size, and benefit from large, cleanly segmented objects. On the other hand, the emergent distribution in MOVi-C (Fig. 15) and COCO (Fig. 16) weights edge artifacts more. We posit that since the object annotations of these datasets contain object positions in the image, edge objects tend to be more discriminative.

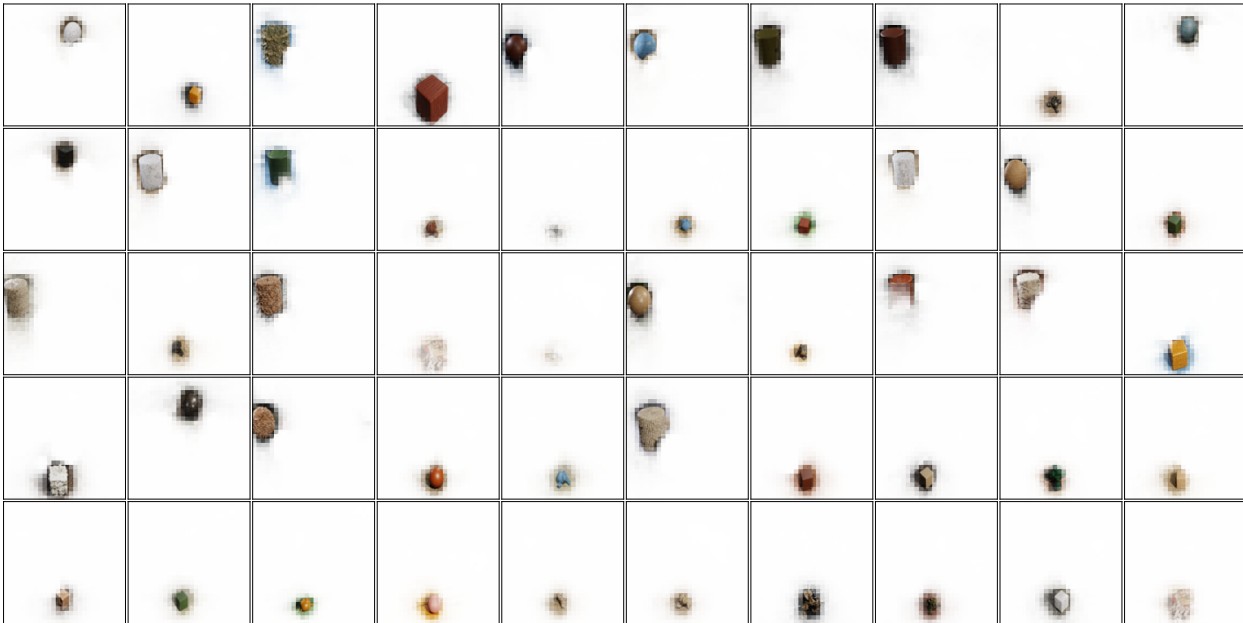

Figure 14: Top 50 slots (left to right, top to bottom) ranked by magnitude from test split of the CLEVrTex dataset.

### D.1.2   Top Learned Primitives

Figs. 17-19 show the t-SNE visualization (van der Maaten & Hinton, 2008) of representations learned by the schema encoder on the schema primitives. In each plot, primitives in the test split are encoded context-free. The learned embeddings exhibit clustering effects on property categories.

### D.1.3   Searching over Slots

We found that the NSI metric, which learns to match entire schemas to entire images, can also be used zero-shot to reliably retrieve individual slots from primitives. We demonstrate two instances of slot search in Figs. 20 and 21. A query in the form of a single primitive is embedded using the schema encoder. The query embedding ranks the slot of a database formed from the test split. In Figs. 20 and 21, we show the property search and position search results, respectively.

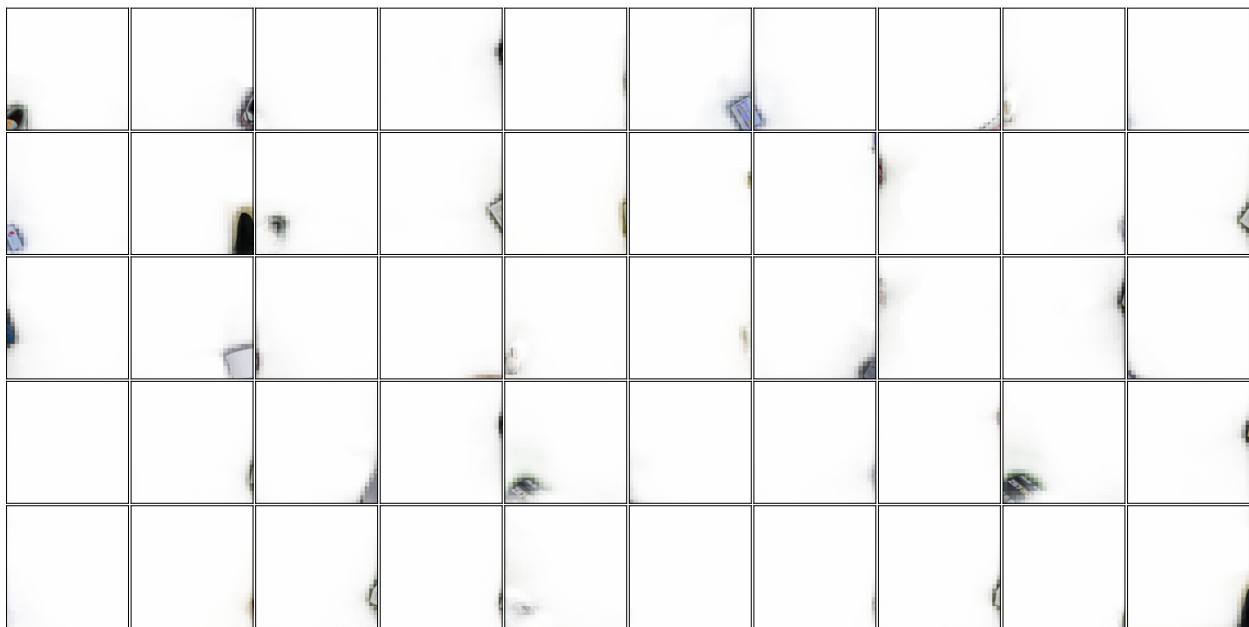

Figure 15: Top 50 slots (left to right, top to bottom) ranked by magnitude from test split of the MOVi-C dataset.

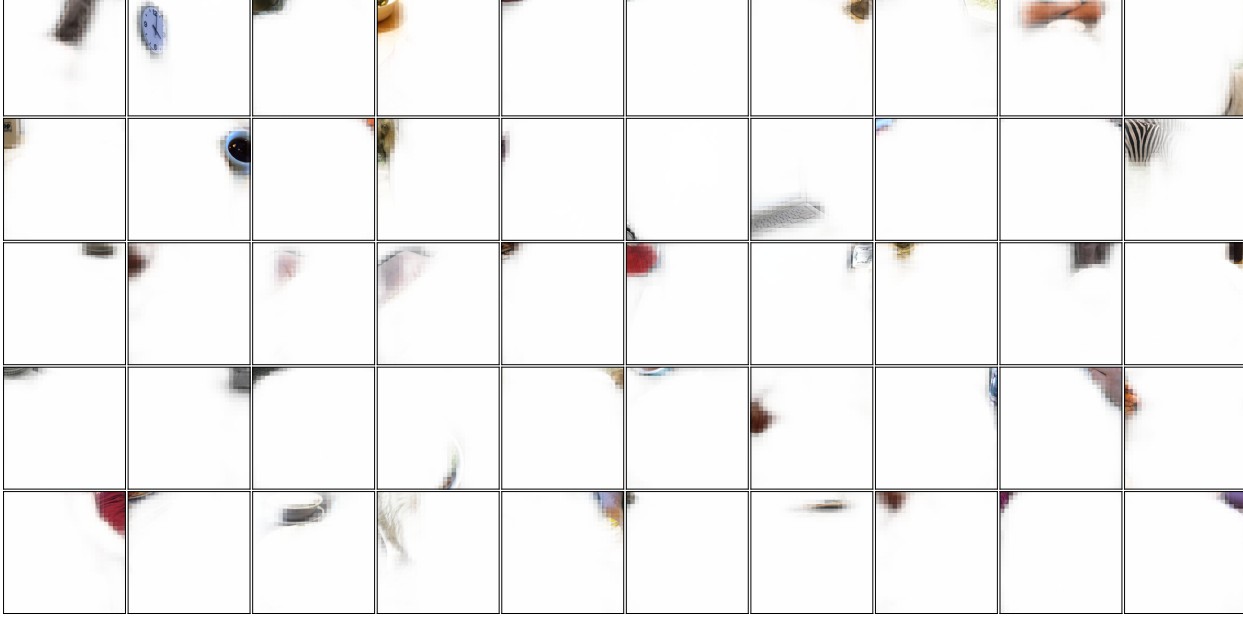

Figure 16: Top 50 slots (left to right, top to bottom) ranked by magnitude from test split of the COCO dataset.

### D.1.4   Slot Sweep for COCO Retrieval Task

Fig. 22 shows the effect of the number of slots on the recall rates. Recall is highest at 15-20 slots for COCO scenes and the model overfits as the number of slots increases further.

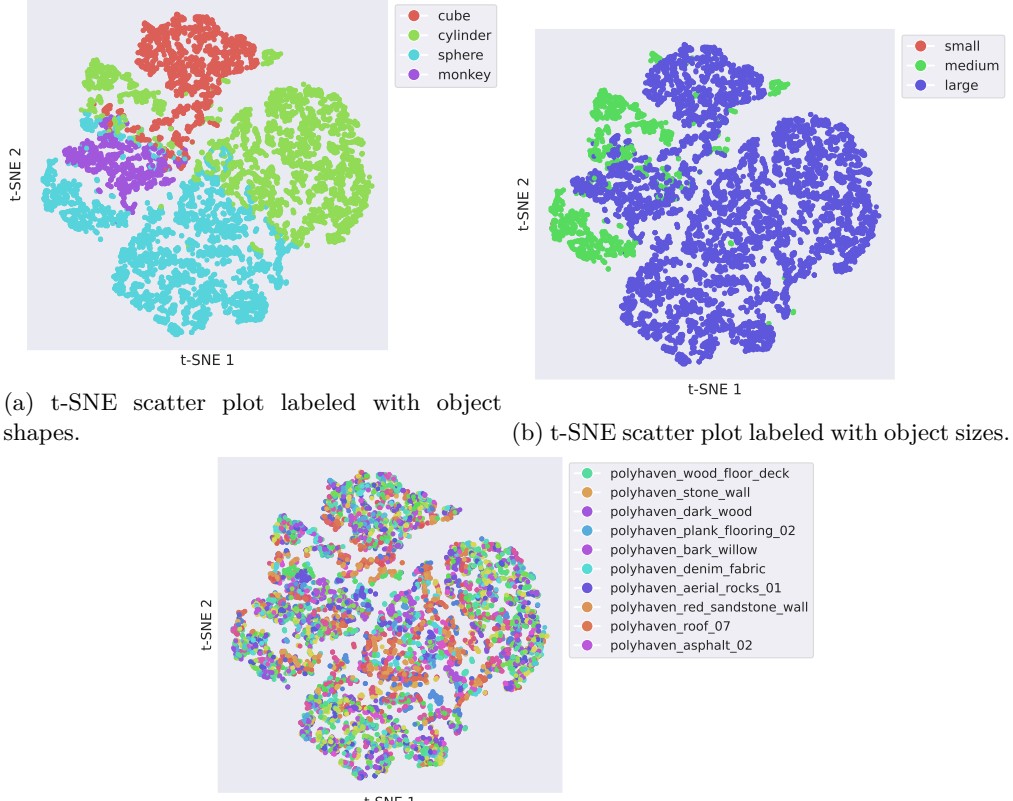

(a) t-SNE scatter plot labeled with object shapes.

(b) t-SNE scatter plot labeled with object sizes.

(c) t-SNE scatter plot labeled with object materials.

Figure 17: t-SNE scatter plots of the top 10000 CLEVrTex primitives weighted by the $l_2$ norm of their embeddings. The embeddings are clearly clustered by the shape type. Interestingly, when looking at the object size, only large and medium-sized objects are represented in the top primitives.

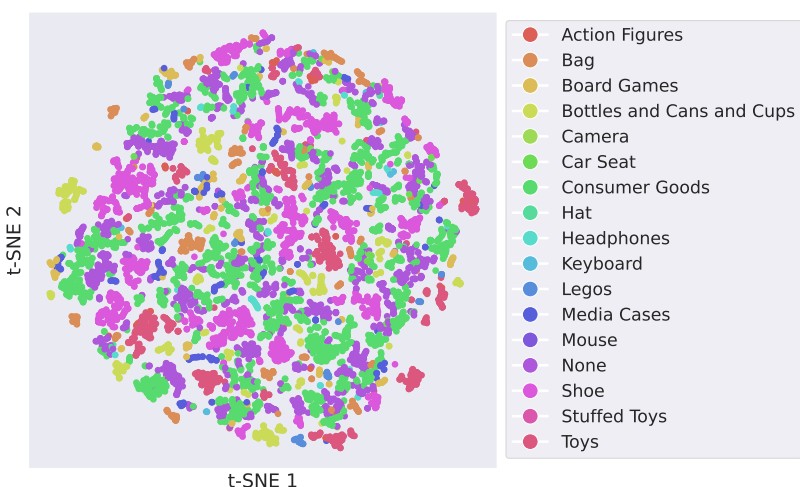

Figure 18: t-SNE scatter plots of the top 10000 MOVi-C primitives weighted by the $l_2$ norm of their embeddings. The scatter points are labeled with the object categories. While there are multiple local clusters, a larger clustering effect based on object categories are not apparent.

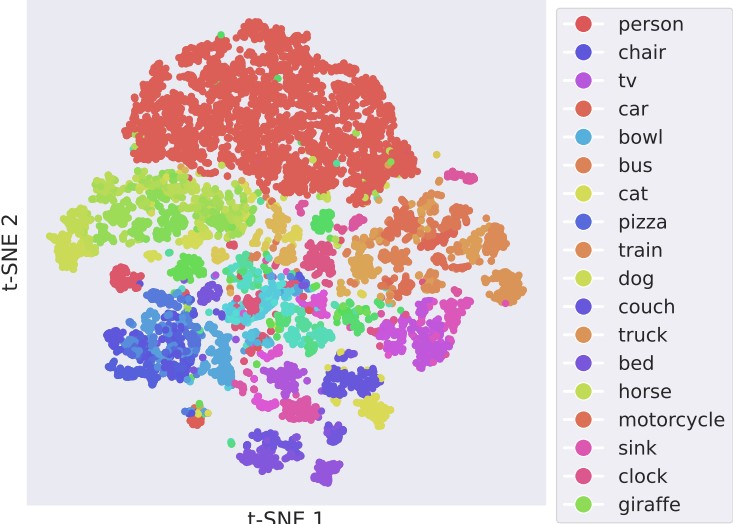

Figure 19: t-SNE scatter plots of the top 10000 COCO primitives weighted by the $l_2$ norm of their embeddings. The scatter points are labeled with the object categories. A category-based clustering pattern is emergent.

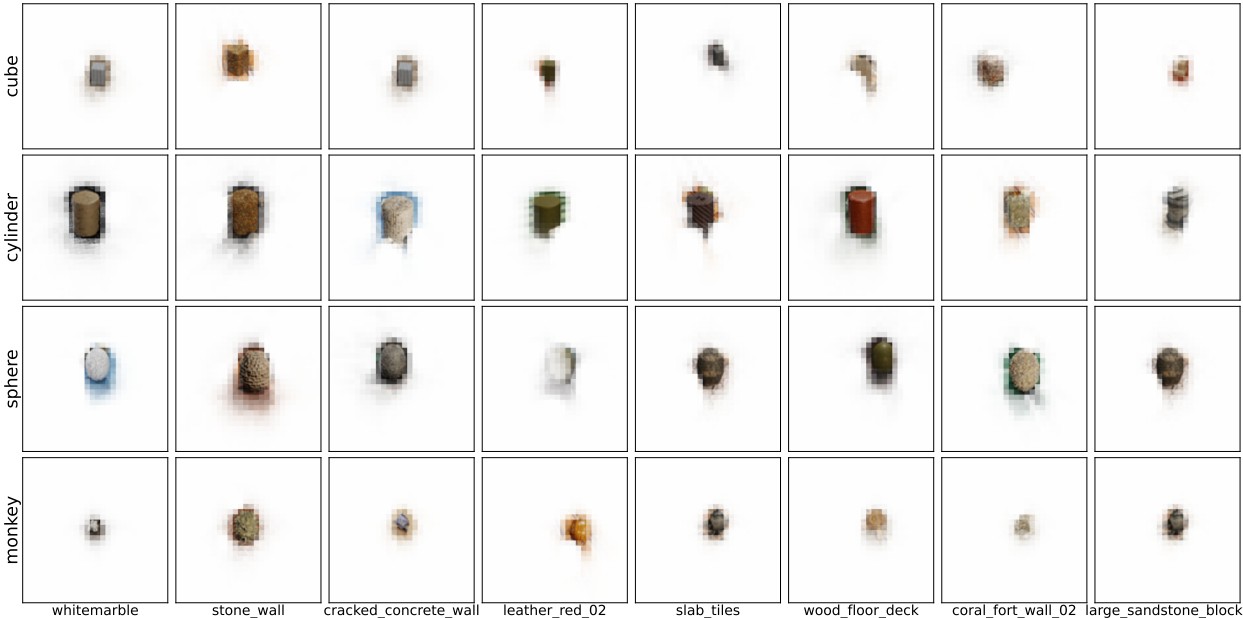

Figure 20: Searching over slots with property queries. The rows represent a shape and the columns represent an object material. The position is fixed at $[0, 0, 0]$ and the size is set to 'Large.' The alignment model reliably retrieves slots with desired object properties if they exist in the database.

### D.1.5 Retrieval with Underspecified Annotations

We investigated the efficacy of grounding under the setting where the annotations of scenes were underspecified. Figs. 23(a) and (b) show the retrieval results under two settings: (a) a maximum of ten objects annotated in a scene and (b) a maximum of five objects annotated in a scene. First, we observed that the drop in recall rates for the ResNet backbone was significant compared to the full setting. Second, CLIP and NSI were resilient to the limited annotation, with NSI outperforming other methods. CLIP benefited

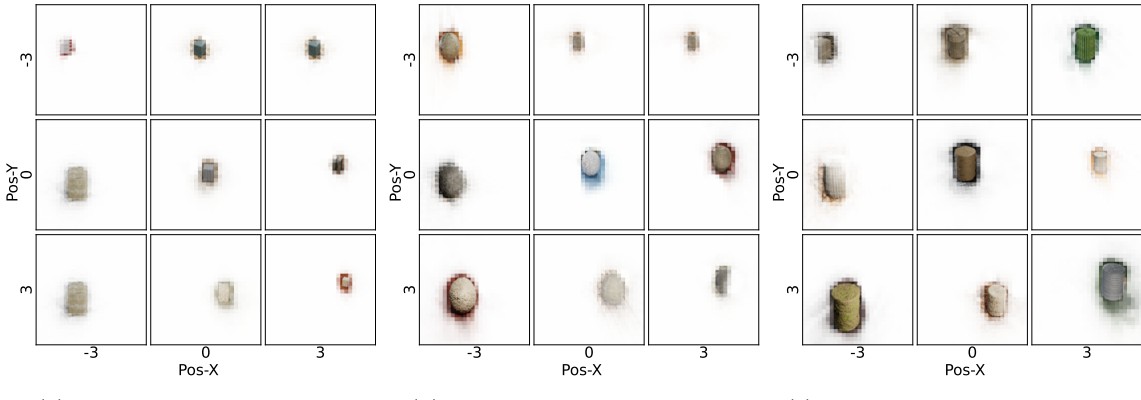

(a) Position search on cubes.   (b) Position search on spheres.   (c) Position search on cylinders.

Figure 21: Searching over slots with position queries. We adjust the 'Pos-X' for each shape on the $x$-axis and 'Pos-Y' on the $y$-axis. The top-ranked slot clearly reflects the position adjustment.

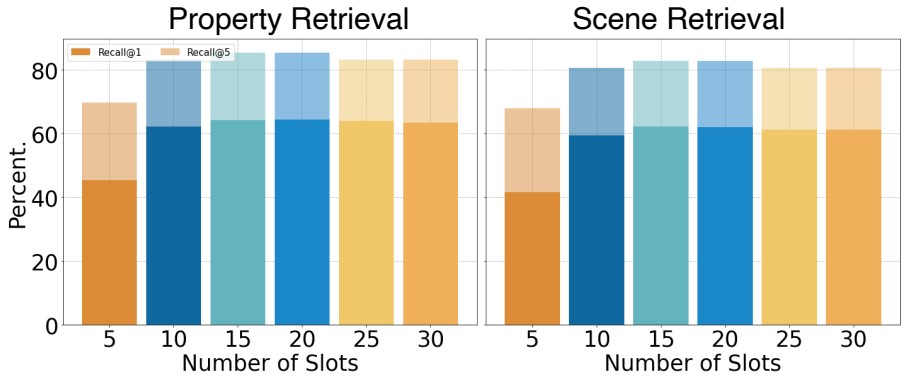

Figure 22: Ablation on the number of slots for scene-property retrieval task. The standard deviation over five seeds was $< 0.3$ across datasets.

from the vast training corpora that helped it generalize to the underspecification. On the other hand, the compositional grounding with strong backbones endows NSI with strong retrieval despite being trained on limited data.

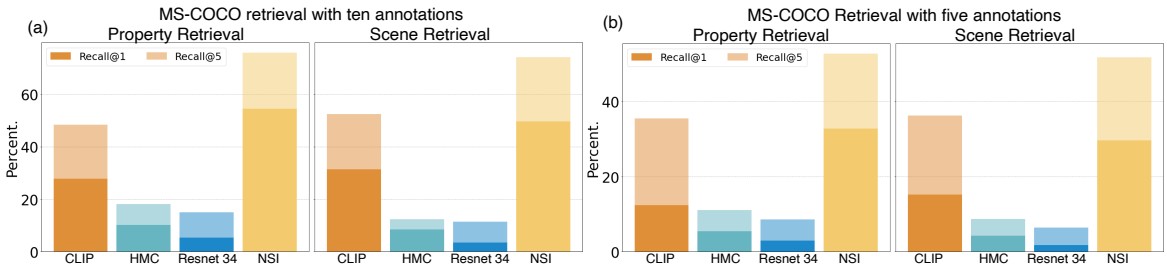

Figure 23: Retrieval results on underspecified scene annotations settings.

### D.1.6 Additional Qualitative Results

Fig. 24 shows additional results on the associations inferred by NSI on MOVi-C and MS COCO 2017.

### D.1.7   Retrieval Results across Property Types

We trained and compared the MOVi-C scene retrieval via NSI and BBox QKV on two separate tasks – one where schemas only contain non-positional object properties (category, size) and one where schemas only contain object positions (center coordinates, bounding boxes). The results are shown in Tab. 6. First, we found that retrieval results across both models and both instances are worse off compared to the completely populated schemas. However, we observe that BBOx QKV relies on positional properties more than non-positional semantics for retrieval and even outperforms NSI on the former type. On the other hand, NSI depends more on non-positional properties and scores significantly higher on retrieving them.

| Model | Recall@5 (%) Non Positional Properties | Recall@5 (%) Positional Properties |
|---|---|---|
| NSI | 58.51 | 43.73 |
| BBox QKV | 38.25 | 45.53 |

Table 6: MOVi-C scene retrieval results on schemas populated by only specific object property types

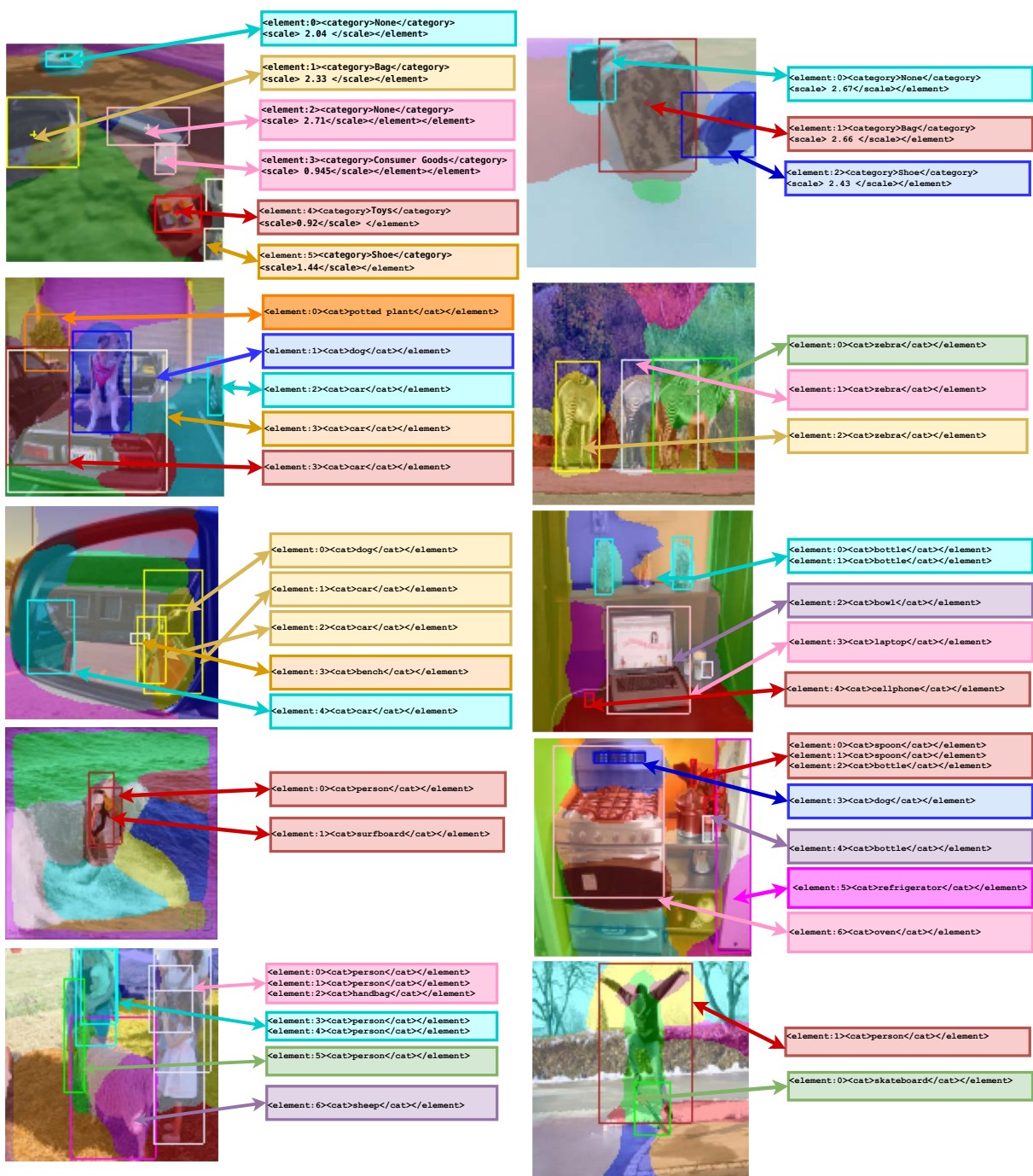

Figure 24: Correspondences inferred by NSI on MOVi-C and COCO scenes.

### D.2    Object Discovery

### D.2.1    Object Discovery Results

| Dataset | Metric | Ungrounded | HMC Matching | NSI |
|---|---|---|---|---|
| CLEVrTex | FG-ARI | $87.79 \pm 0.12$ | $88.37 \pm 0.12$ | $89.89 \pm 0.01$ |
| | mBO | $44.86 \pm 0.04$ | $45.23 \pm 0.23$ | $46.60 \pm 0.02$ |
| MOVi-C | FG-ARI | $65.53 \pm 0.15$ | $65.61 \pm 0.31$ | $66.41 \pm 0.12$ |
| | mBO | $36.79 \pm 0.03$ | $36.79 \pm 0.41$ | $38.52 \pm 0.23$ |
| COCO | FG-ARI | $40.12 \pm 0.29$ | $32.18 \pm 0.45$ | $44.24 \pm 0.27$ |
| | $\text{mBO}^i$ | $27.20 \pm 0.31$ | $18.32 \pm 0.51$ | $28.12 \pm 0.25$ |
| | $\text{mBO}^c$ | $26.54 \pm 0.25$ | $19.61 \pm 0.82$ | $32.10 \pm 0.31$ |
| PASCAL VOC 2012 (Zero-Shot) | FG-ARI | $20.42 \pm 0.13$ | $15.14 \pm 0.09$ | $21.97 \pm 0.17$ |
| | $\text{mBO}^i$ | $35.97 \pm 0.15$ | $25.15 \pm 0.08$ | $36.98 \pm 0.19$ |
| | $\text{mBO}^c$ | $37.94 \pm 0.17$ | $27.02 \pm 0.12$ | $39.06 \pm 0.22$ |

Table 7: Object discovery results. We use the DINO backbone and an MLP decoder across methods. The standard deviation was calculated over five random seeds.

Table 7 contains the complete set of object discovery results with the standard deviations. We also conducted a zero-shot evaluation of the models trained on MS-COCO on Pascal VOC 2012 (Everingham et al., 2015). We observed that the benefits of grounding the model via NSI on one real-world dataset were transferred to the other for object discovery, with the grounded model also improving mask segmentation over its ungrounded counterparts for Pascal VOC.

### D.2.2    Slot Sweep over COCO Scenes

Fig. 25 shows the NSI object discovery results on COCO scenes as the number of slots is increased. We observe that segmentation improves until 15 slots and then slowly tapers off.

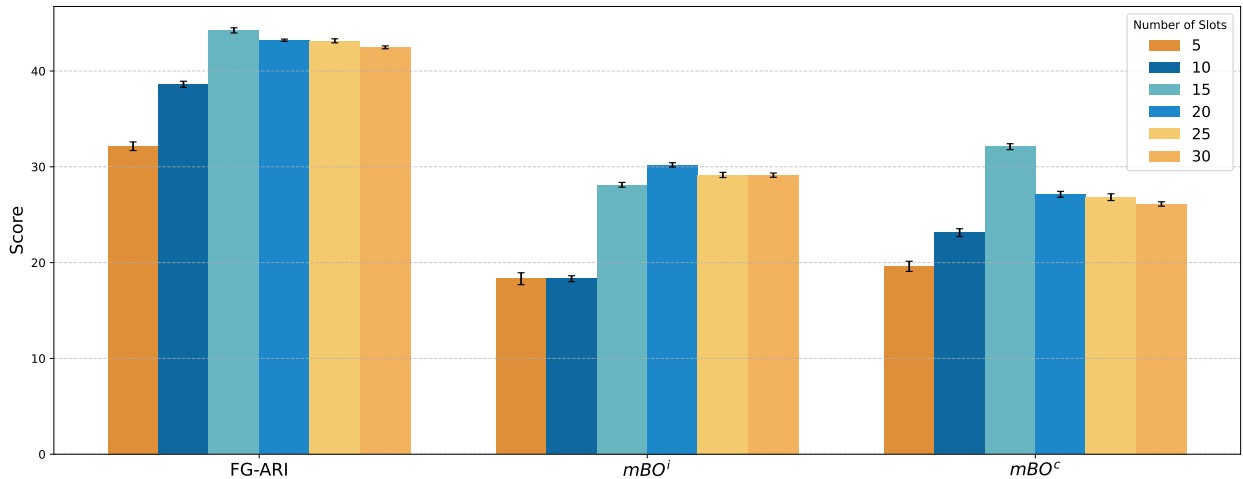

Figure 25: Ablation on the number of slots for COCO object discovery using NSI.

### D.2.3    Property Ablations on COCO Scenes

In Fig. 26, we ablate the properties used to form schema primitives. Ostensibly, the slots weigh bounding box coordinates more than categories, as the performance drop is steeper when the former is ablated as opposed to the meager loss when the latter is ablated.

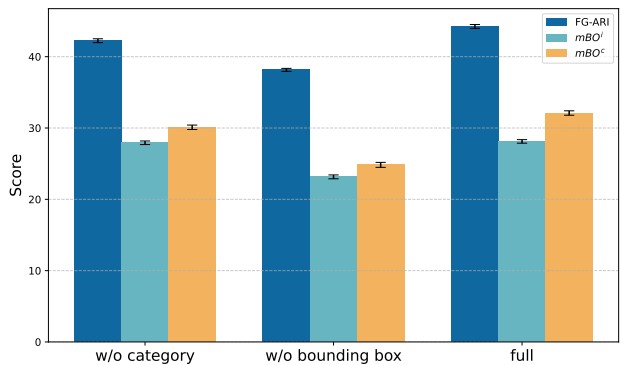

Figure 26: Ablation on the schema properties for NSI.

### D.2.4 Grounding Examples Ablation on COCO Scenes

Fig. 27 shows the results of ablating over the number of grounding examples used for NSI co-training. We observe that the *ARI* metric and the class-wise *mBO* are sensitive to grounding, with as few as 100 annotations improving object discovery via segmentation masks.

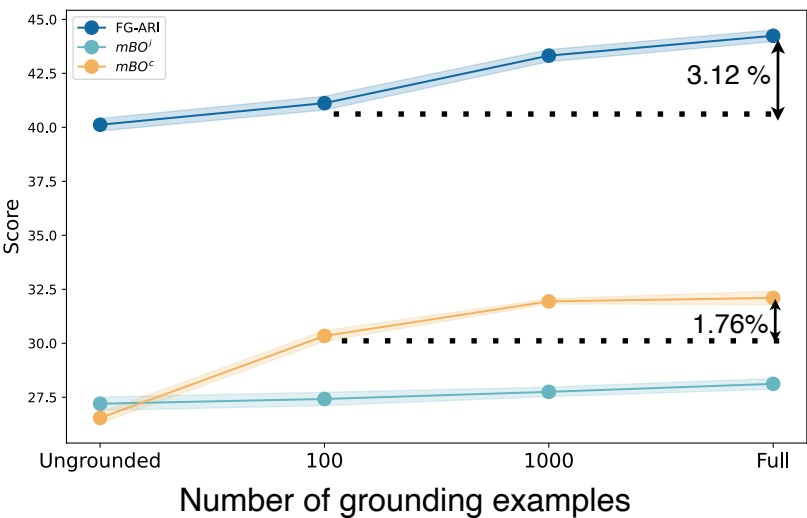

Figure 27: Ablation on number of grounding examples used to train NSI on MS COCO.

### D.2.5 Mask Visualizations

Figs. 28, 29, 30, 31 visualize slot masks over scenes from CLEVrTex, MOVi-C, COCO, and Pascal VOC, respectively.

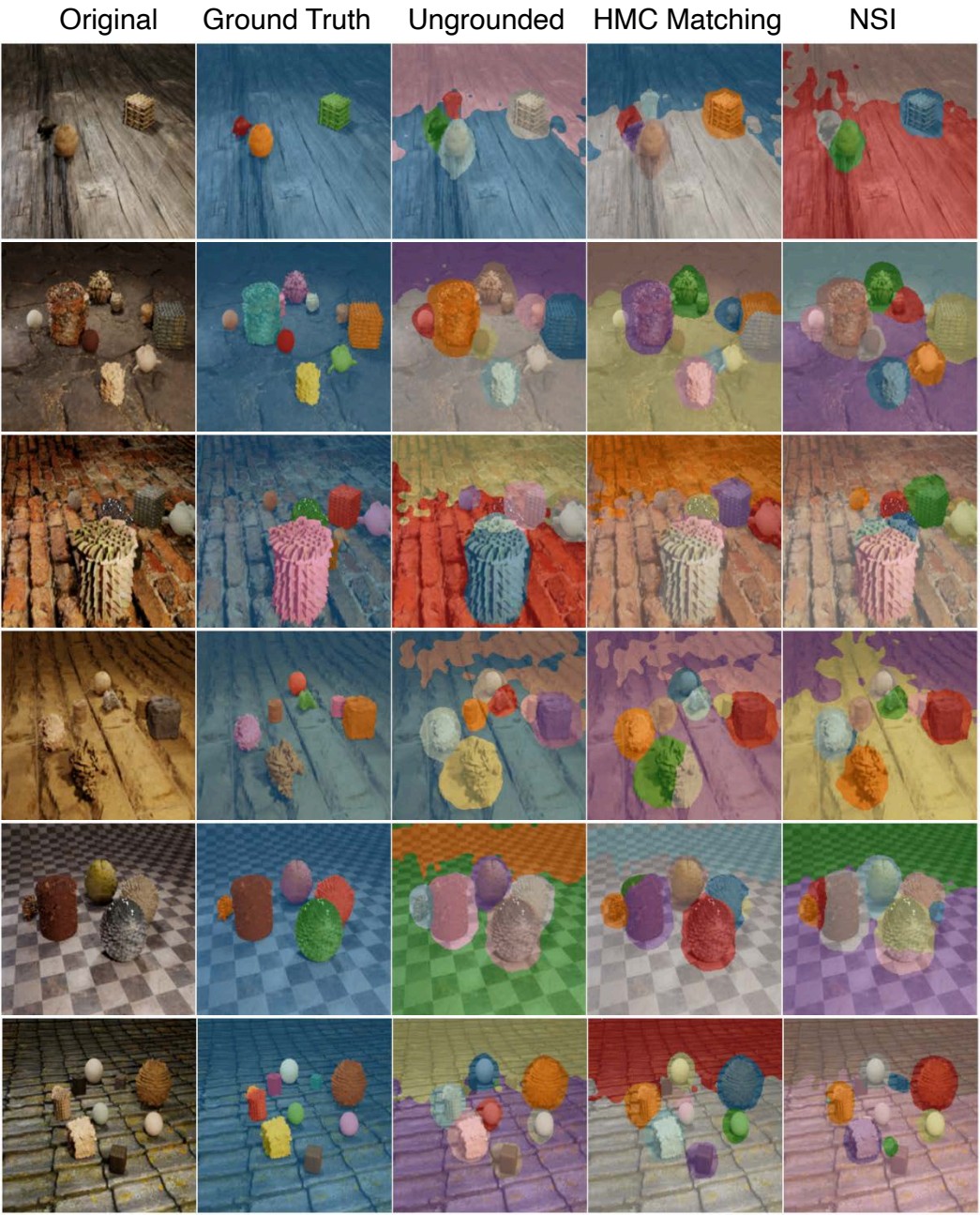

Figure 28: Object discovery results on CLEVrTex scenes.

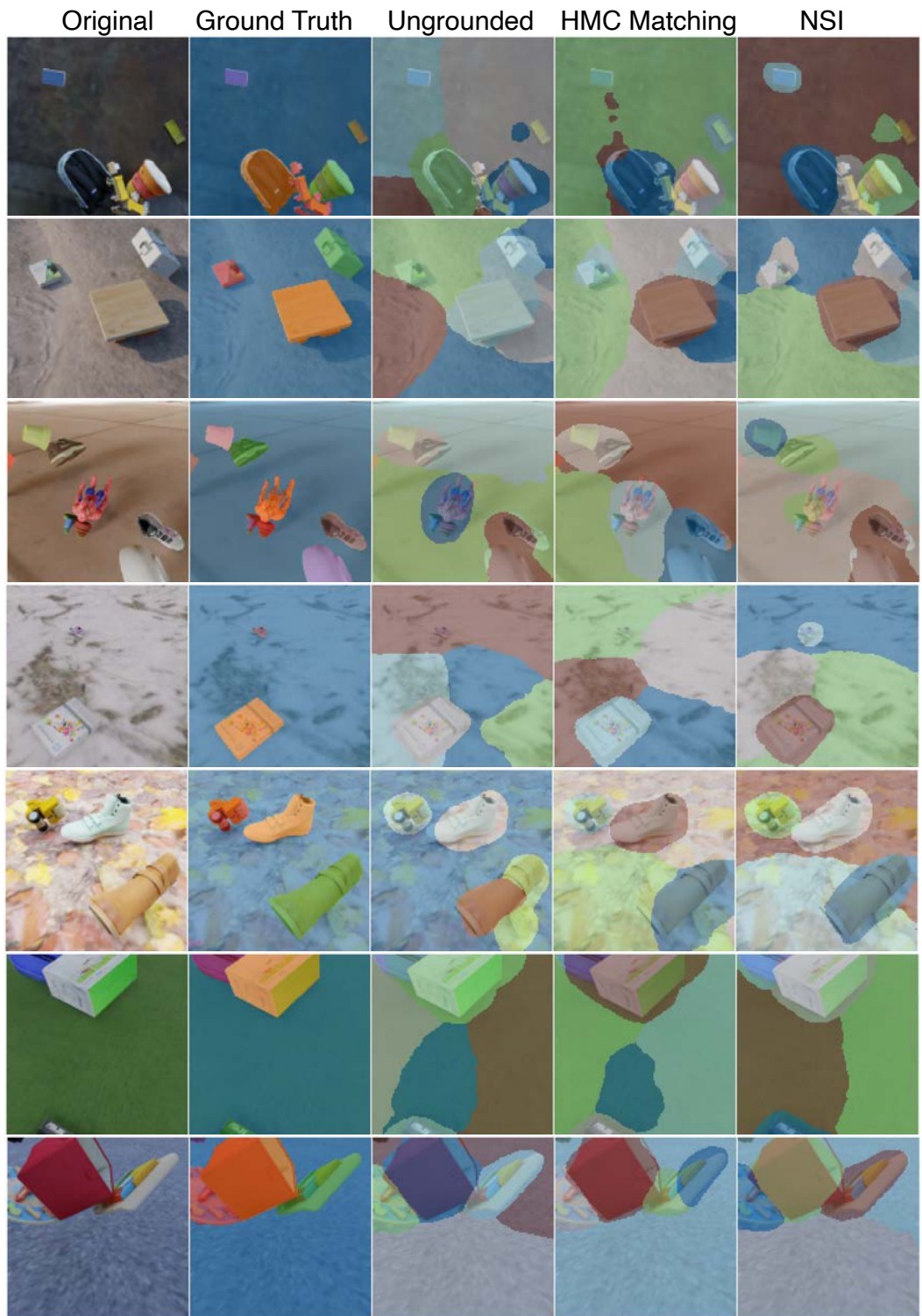

Figure 29: Object discovery results on MOVi-C scenes.

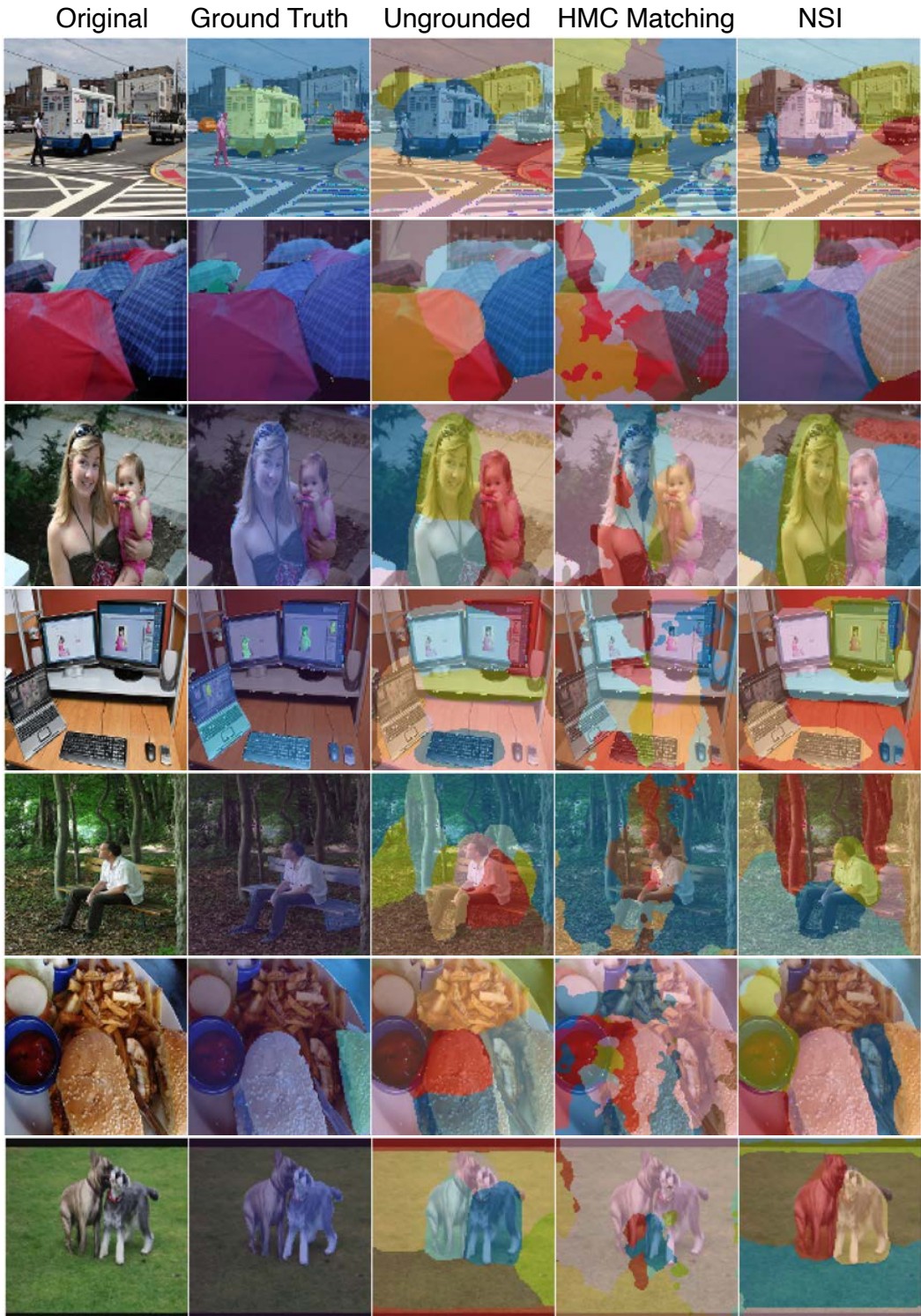

Figure 30: Object discovery results on COCO scenes.

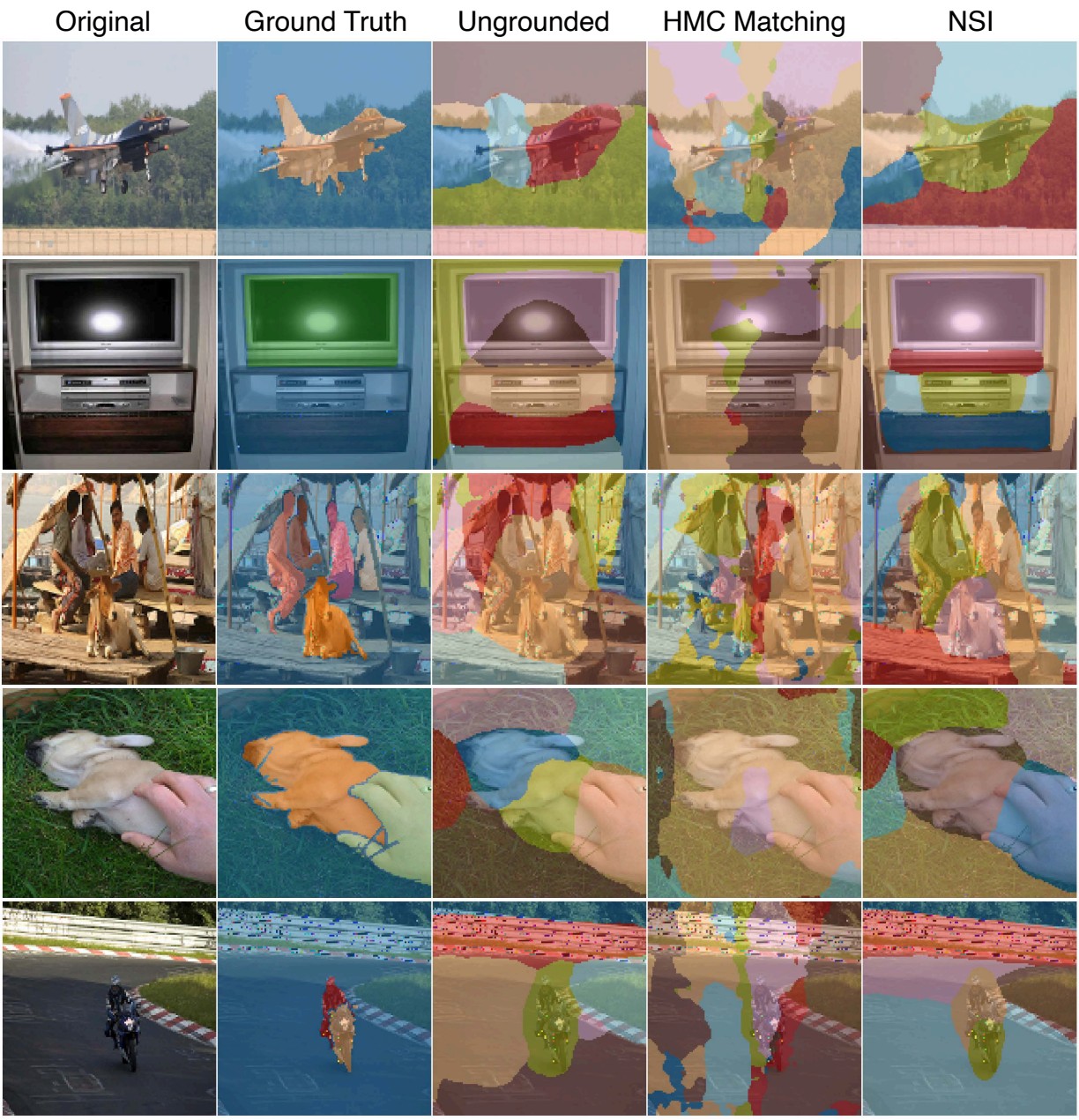

Figure 31: Zero-shot object discovery results on Pascal VOC 2012 scenes.

### D.3 Grounded Slots as Visual Tokens

### D.3.1 Confusion Matrices

Figs. 32 and 33 show the classification confusion matrices for the CLEVr-Hans 3 and CLEVr-Hans 7 tasks. In the few-shot setting, the model often predicts scenes into under-specified and less discriminative classes, like "cyan object in front of two red objects." Similarly, it also tends to mispredict into classes containing large objects like "large cube and large cylinder" that contain reasoning over larger objects and are easier to tokenize.

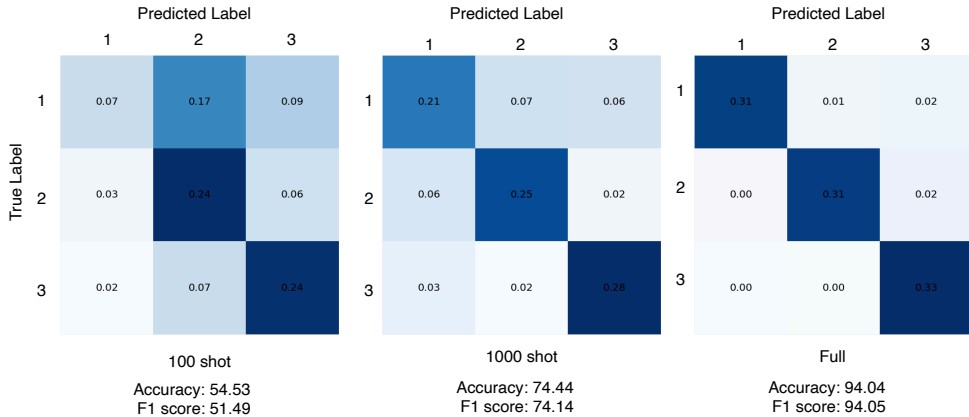

Figure 32: Confusion matrices for NSI prediction on the CLEVr-Hans 3 task.

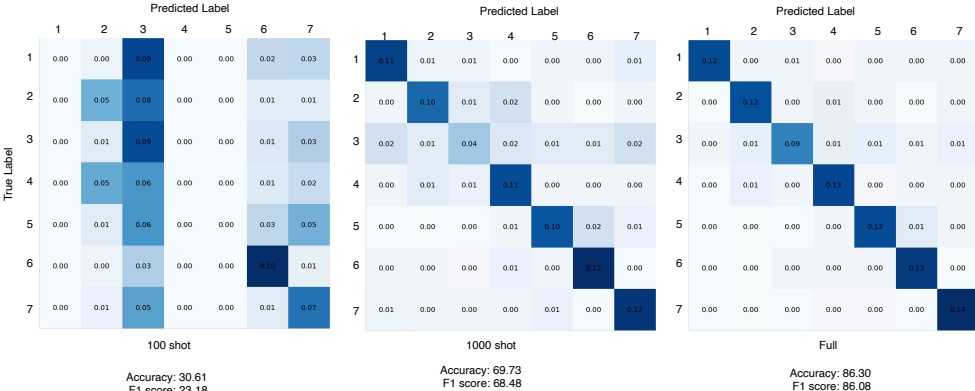

Figure 33: Confusion matrices for NSI prediction on the CLEVr-Hans 7 task.

### D.3.2 Visual Rationales

Fig. 34 simultaneously visualizes the rationales across data settings and attention layers. On an average, we observe that rationales tend to get stronger and more accurate as the number of training examples increases and the ViT depth increases.

Figs. 35 and 36 demonstrate rationales across the grounding continuum for Hans 3 and Hans 7 classification tasks, respectively.

Fig. 37 demonstrates attention maps where the model prediction is correct, but the rationale is not entirely dispositive.

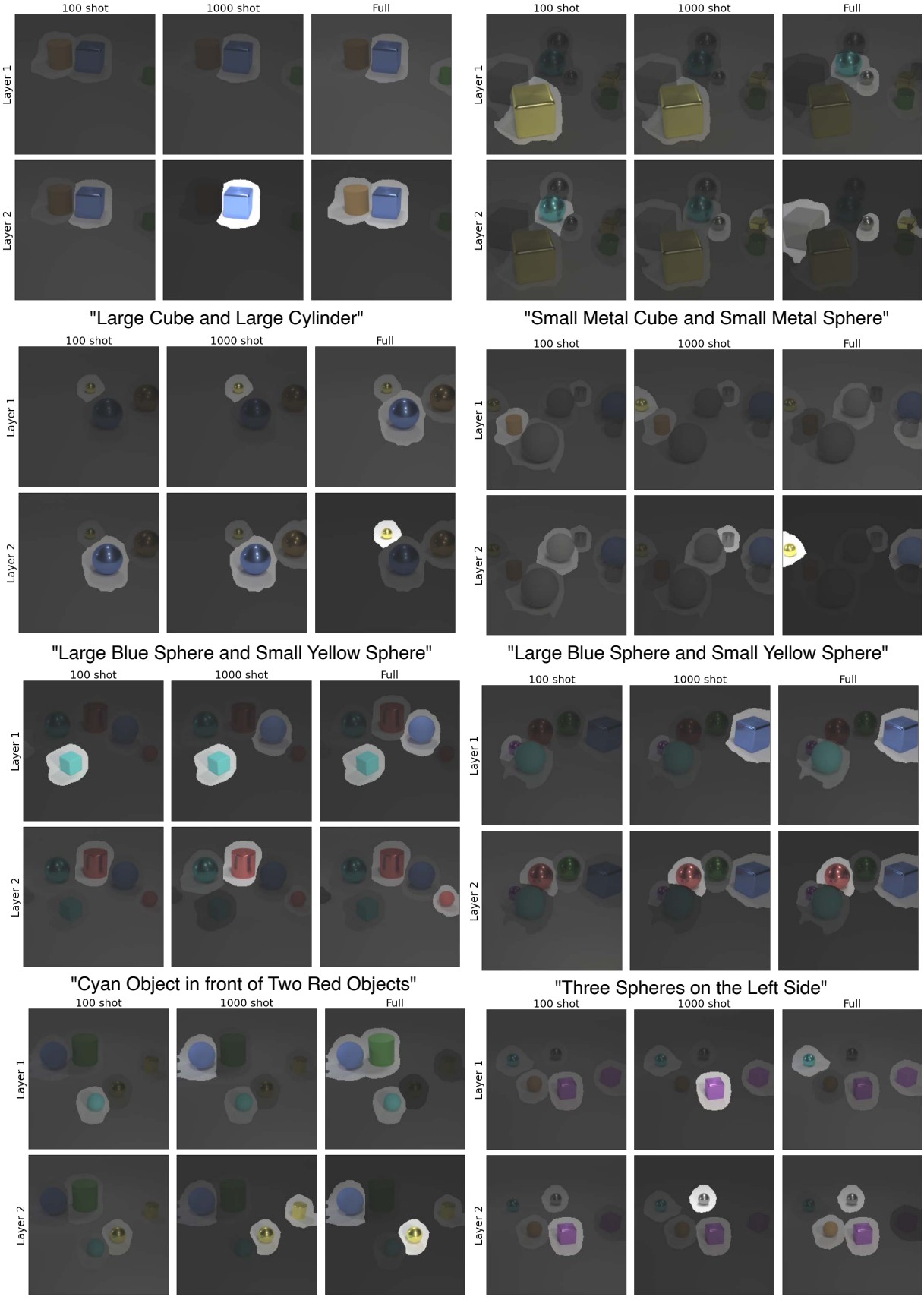

Figure 34: NSI visual rationales across different data regimes and attention depth.

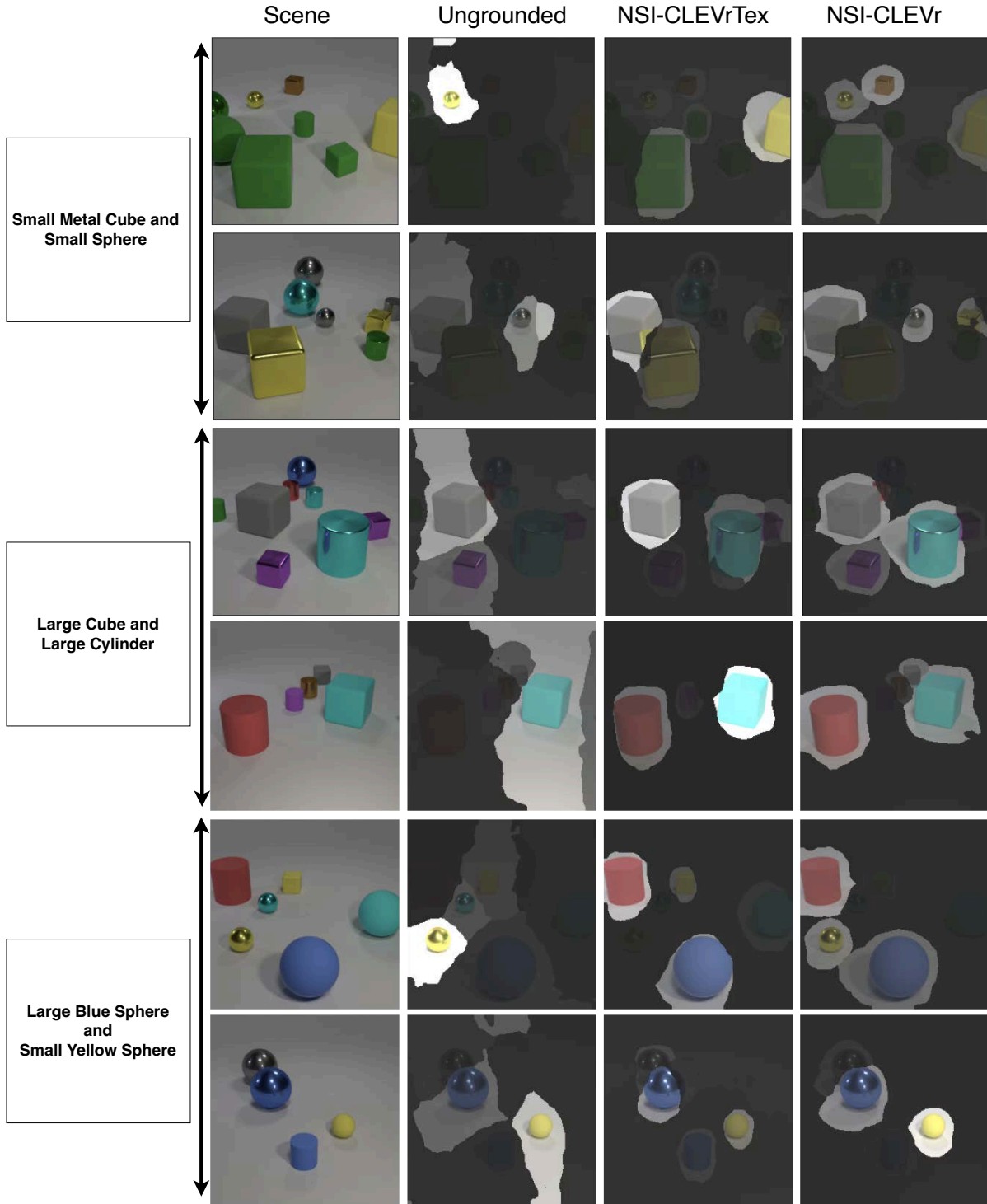

Figure 35: Visual rationales on Hans 3 across different methods.

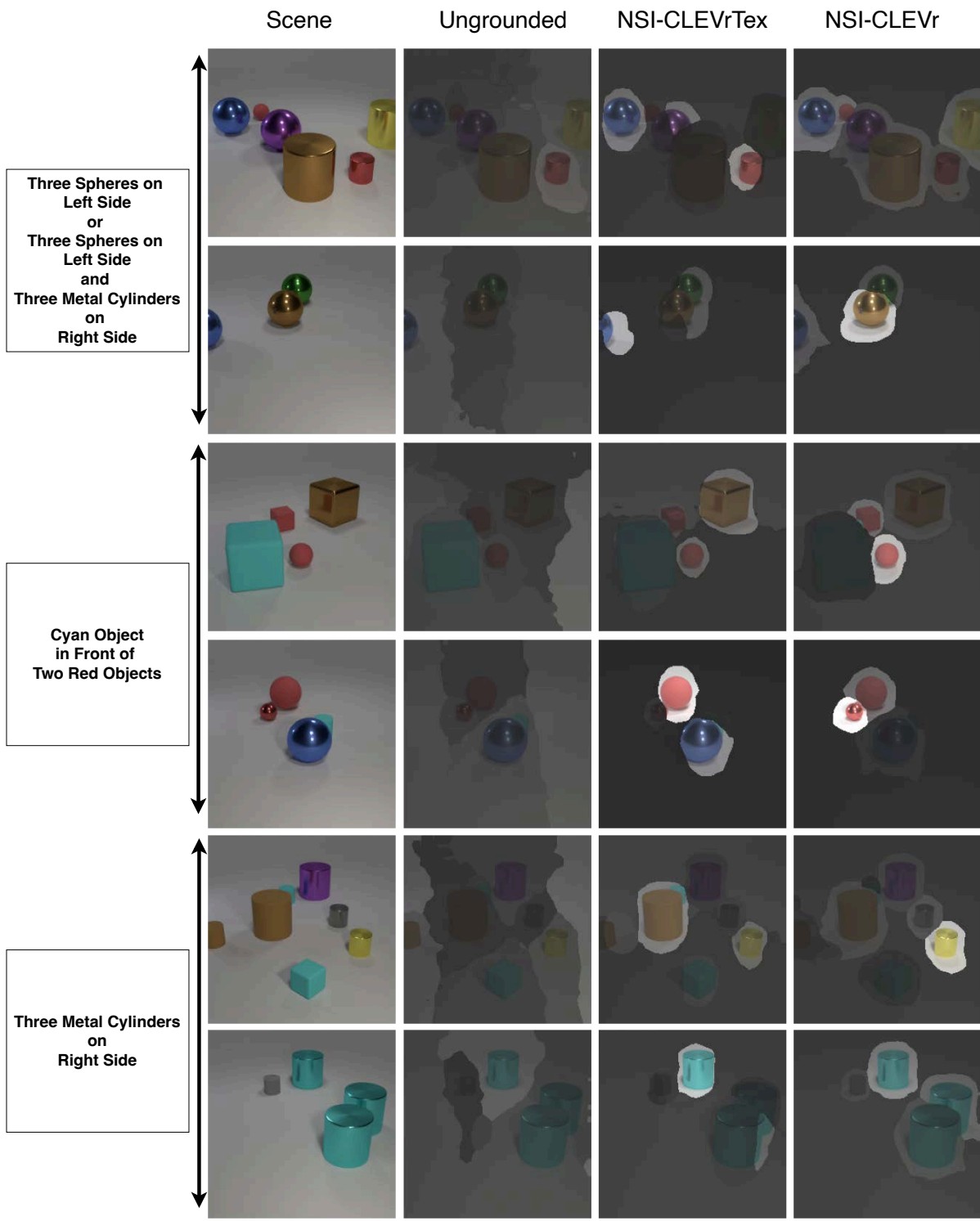

Figure 36: Visual rationales on Hans 7 across different methods.

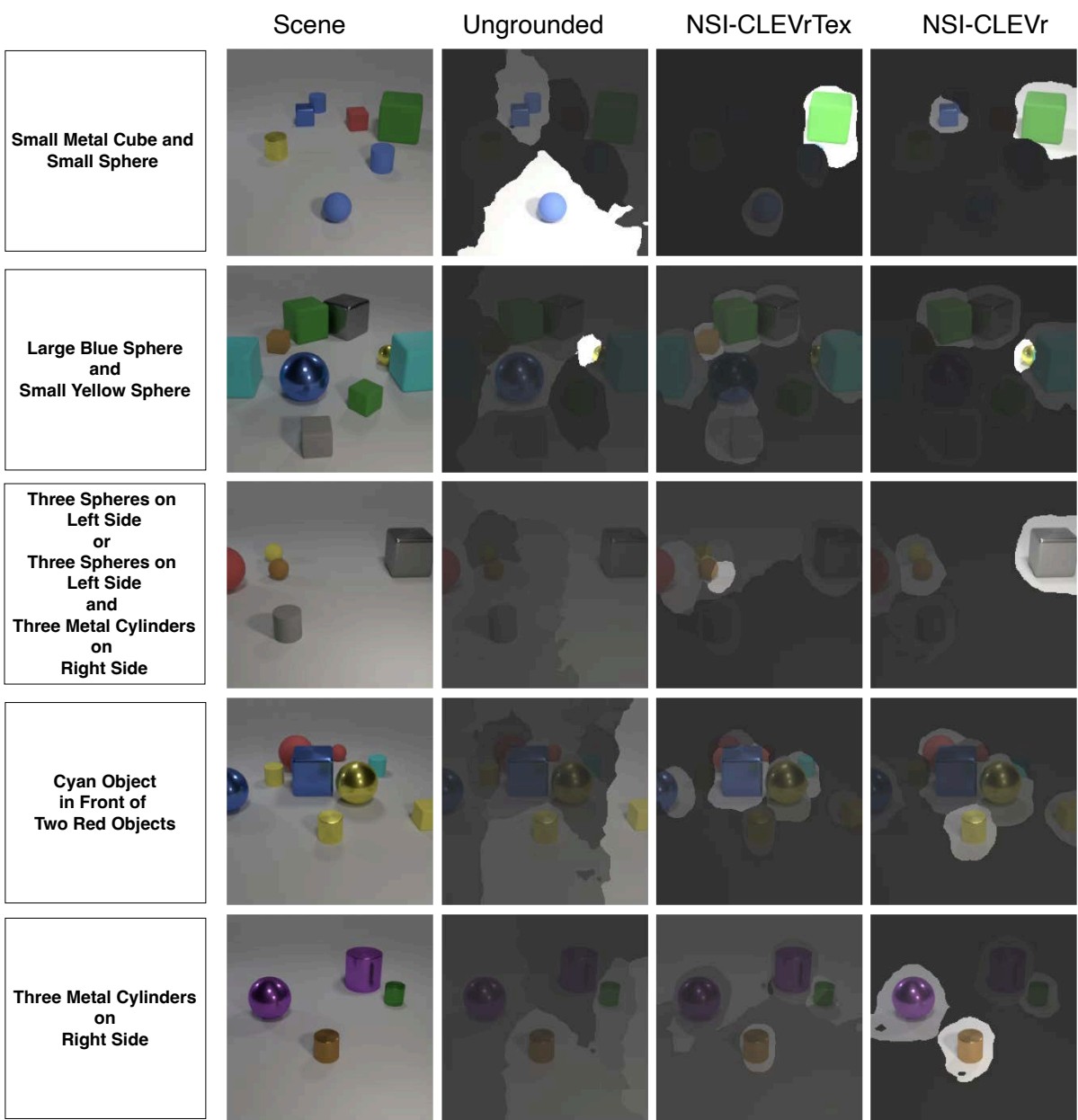

Figure 37: Some failure cases where NSI makes the correct prediction, albeit using incorrect or ambiguous support slots.

# E   Object Detection with NSI

In this section, we demonstrate an architecture that uses inferred correspondences by NSI to perform object detection. We run preliminary experiments and explore the use of slots in real-world visual reasoning systems.

## E.1   NSI Schema Generator Model

We formulate the NSI schema generator (see Fig. 38) to predict, from each slot, its corresponding schema primitives. To this end, we modify an encoder Transformer by interleaving cross-attention blocks with the self-attention layers to assimilate the context from the encoded slots. The input to the model is simply learned positional embeddings $\mathcal{P}^{1:N}$ that are decoded by $L$ attention-ensemble stacks to output primitive representations. We call this architecture $SET_\theta(.)$. The Slot Encoder Transformer (SET) is a stack of $L$ blocks, each computing (a) self-attention over inputs followed by (b) cross-attention of inputs over the context slot. Let $Z^{dec,1:T}_{prim,l-1}$ be the sequence representation at layer $l-1$ and $S$ denote the slot. Then

$$Z^{dec,1:N}_{prim,L} = SET_\theta(\mathcal{P}^{1:N}, S) \tag{18}$$

The cross-attention implementation of block $l$ is shown in Algorithm 2.

---

**Algorithm 2** Cross-Attention for Block $l$ of SET

---

**Require:** $Z^{dec,1:T}_{prim,l} \in \mathbb{R}^{T \times d}$, $T$ primitive embeddings from self-attention of layer $l$

**Require:** $\mathcal{S} \in \mathbb{R}^d$, Context Slot

Get query tokens: $Q^{1:T}_{l-1} = MLP_Q(Z^{dec,1:T}_{prim,l})$

Get keys, values of slot $\mathcal{S}$: $K, V = MLP_{KV}(S)$

Compute attention values: $M = \text{softmax}(Q^T K / \sqrt{d})$

Get output: $Z^{dec,1:T}_{prim,l} = M \times V$

---

Using an encoder-transformer-styled predictor has two advantages: (1) primitives can be generated in parallel and (2) each primitive representation is aware of the overall prediction context. $MLP$ property heads predict the object properties from the $Z^{dec,1:N}_{prim,L}$ representations. At each training iteration, the predicted properties $\hat{p}^{1:N}$ are optimally matched to their ground truth labels $p^{1:N}$ via an ordering $\sigma(1:N)$ obtained from Hungarian matching (HM) (Kuhn, 1955) on the property prediction loss $\mathcal{L}_{properties}$. Note that $\mathcal{L}_{properties}$ is a per-primitive loss obtained from the sum of individual property prediction losses. We use the cross-entropy loss for discrete properties, mean-squared error for continuous properties, and augment bounding-box regression with the Intersection over Union (IoU) loss. The training objective $\mathcal{L}_{gen}(.)$ is formulated as follows:

$$\sigma(1:N) = HM\left(\mathcal{L}_{properties}\left(p^{1:N}, \hat{p}^{1:N}\right)\right) \tag{19}$$

$$\mathcal{L}_{gen} = \sum_{i=1}^{N} \mathcal{L}_{properties}(p_i, \hat{p}_{\sigma(i)}) \tag{20}$$

At training time, we pad the aligned ground-truth instances with no-object labels $\Phi$ to account for representations without object predictions. In practice, bipartite matching for a single-slot instance is more computationally feasible than the overall image instance because per-slot object instances are significantly fewer.

## E.2   NSI Schema Generator Hyperparameters

The hyperparameters for the NSI schema generator are listed in Table 8.

## E.3   Experiments

The correspondences from the train split are used to learn the NSI schema generator, as outlined in Appendix E.1. The schema generator decodes $T$ primitives from each slot, including the confidence level of

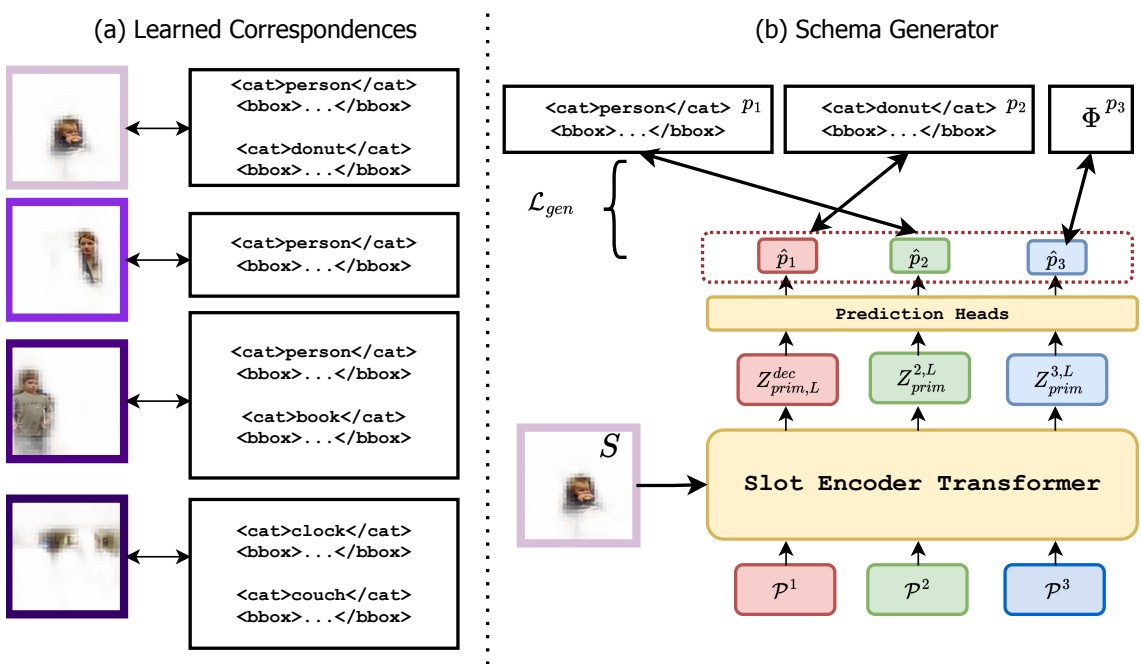

Figure 38: NSI schema generator model: (a) Learned correspondences between slot and schema primitives are used to train the schema generator. (b) A Slot Encoder Transformer attends to individual slots via cross-attention and, in parallel, decodes tokens into schema primitives. At training time, a Hungarian set matching procedure assigns predictions to primitives associated with the slot. The prediction error is aggregated over assignments to compute $\mathcal{L}_{gen}$.

| Module | Hyperparameters | MOVi-C | COCO-10 | COCO-30 |
|---|---|---|---|---|
| SET | Num. Layers | 2 | 2 | 2 |
| | Num. Heads | 2 | 2 | 2 |
| | Hidden Dims. | 192 | 192 | 192 |
| | Predictions per Slot ($T$) | 5 | 8 | 8 |
| Inference | top-$M$ predictions | 10 | 10 | 30 |
| | Non-Max Suppression/Threshold | Yes (0.75) | Yes (0.75) | Yes (0.75) |
| Training Setup | Batch Size | 64 | 64 | 64 |
| | LR Warmup steps | 10000 | 30000 | 30000 |
| | Peak LR | $4 \times 10^{-4}$ | $1 \times 10^{-4}$ | $1 \times 10^{-4}$ |
| | Dropout | 0.1 | 0.1 | 0.1 |
| Training Cost | GPU Usage | 40 GB | 40 GB | 40 GB |
| | Days | 2 | 4 | 4 |

Table 8: Hyperparameters for the NSI schema generator model instantiation and training setup.

each object prediction. The overall schema for a single image is obtained as the top $M$ confident primitives out of predictions from all $K$ slots of that image. The NSI schema generator enables solving of downstream tasks with slots by making use of predicted properties from primitive tags. We demonstrate the usefulness of NSI for object detection.

**Object Detection Results:** In the context of real-world object detection on images, the schema properties can be used to identify and locate objects in diverse scenes from the MOVi-C and COCO datasets. To this end, we extract the (`<cat>`) category and (`<bbox>`) bounding box fields from the generated primitives that depict the object category and location, respectively. For COCO, we test on two variants of the dataset: (1) **COCO-10**, a simple subset of the test split containing ten objects at a maximum, and (2) **COCO-30** that contains as many as 30 objects in a scene. We report the $AP_{IoU}$ metric across all three benchmarks. It denotes the area under the precision-recall curve for a certain IoU threshold (in %). We also run comparisons against the methods outlined in the retrieval experiments. Fig. 39 shows the experimental results and Figs. 40, 41 visualize object detection across various predictors. Our comparison baselines include:

(a) **Vanilla Slot Attention** (Locatello et al., 2020): The model is trained from scratch to resolve slot-object assignments through HMC.
(b) **DINOSAUR** (Locatello et al., 2020): It uses the DINO ViT backbone to learn slots for unsupervised object discovery by reconstructing perceptual features. The architecture is frozen while we train shallow predictors on top to detect objects.
(c) **DINOSAUR-FT**: We use the DINOSAUR model but fine-tune the architecture end-to-end on the prediction task.

We make the following observations:

(a) **NSI outperforms prior set-matching slot predictors**, especially by significant margins (20-30%) at lower IoU thresholds. Grounding object concepts in slots *a priori* improves the predictive power of slots. In comparison, matching the set of slots against the entire set of object annotations of the image yields poor generalization. In addition, large-scale pre-training and end-to-end fine-tuning are crucial ingredients, as evidenced by the subpar performance of the Vanilla and DINOSAUR methods on COCO.
(b) **The performance disparity between NSI and DINOSAUR-FT widens** as the complexity of scenes increases from COCO-10 to COCO-30. The inability of DINOSAUR to predict more than one object per slot necessitates modeling and learning to match up to 30 different slots, which generalizes poorly on novel scenes.
(c) **HMC slots deteriorate** for COCO-30 where the backbone is tasked with matching with 30 different objects at a time.

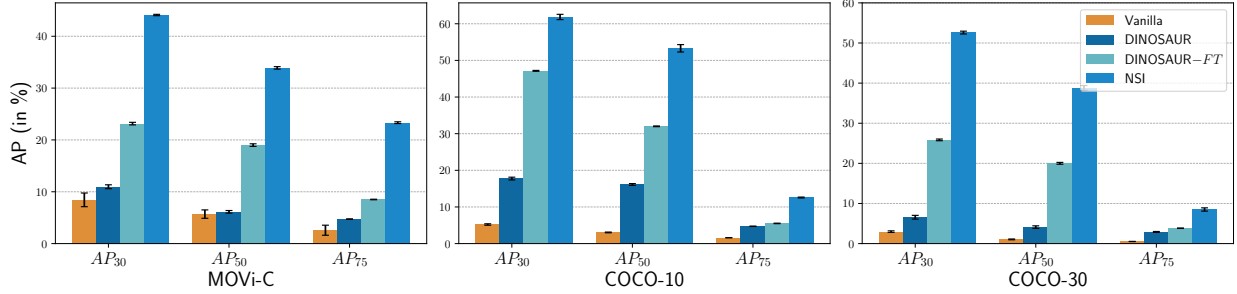

Figure 39: Object detection performance of various prediction methods on the MOVi-C, COCO-10, and COCO-30 benchmarks. We report $AP@IoU$ (higher is better) for different IoU thresholds. Standard deviation is reported over five random seeds.

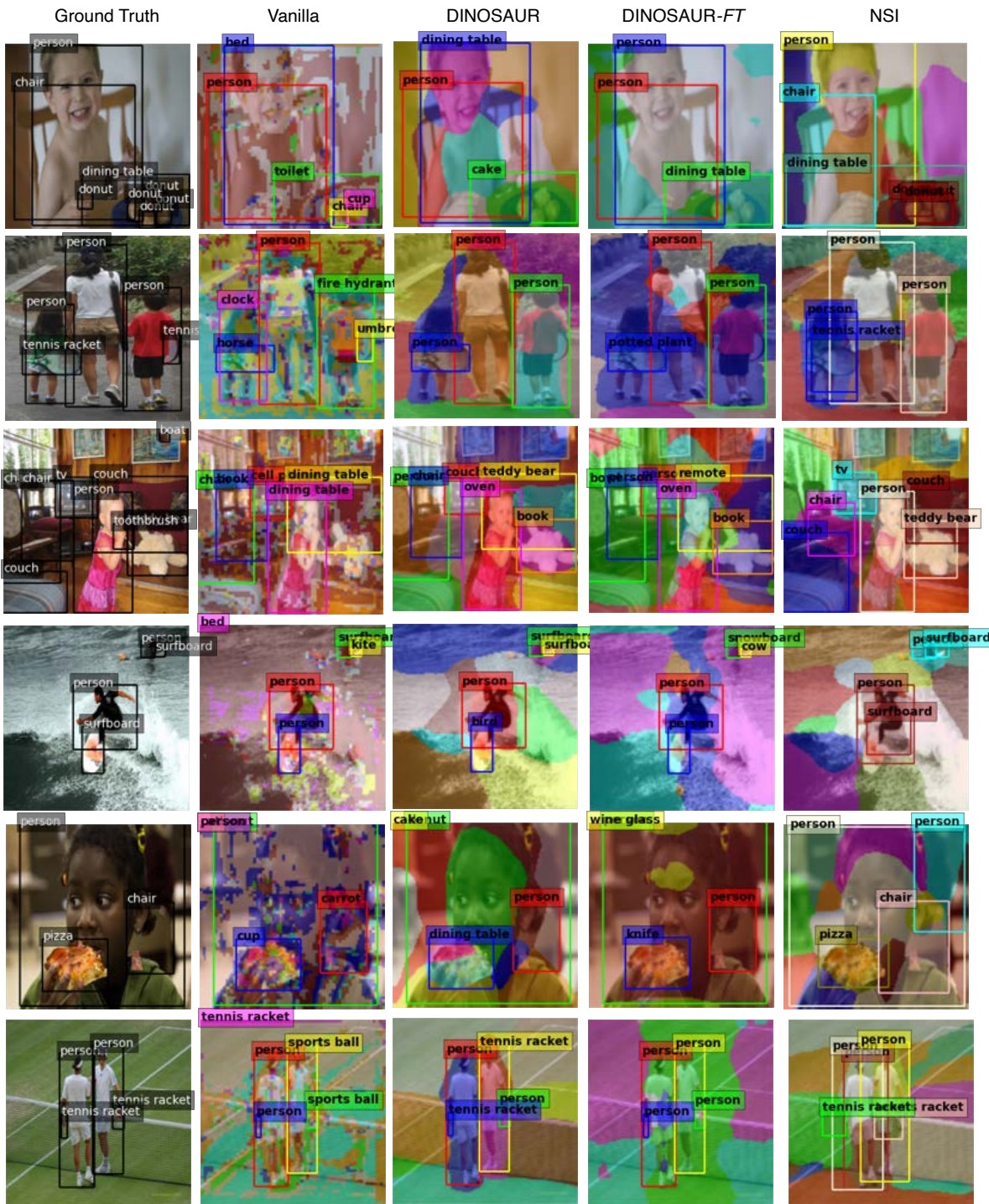

Figure 40: Object detection results on COCO scenes. NSI schema generator can flexibly detect multiple objects from the same slot, as evidenced by detections on COCO images. For example, a single slot predicts multiple 'donuts,' 'chair and dining table,' and 'person and tennis racket'.

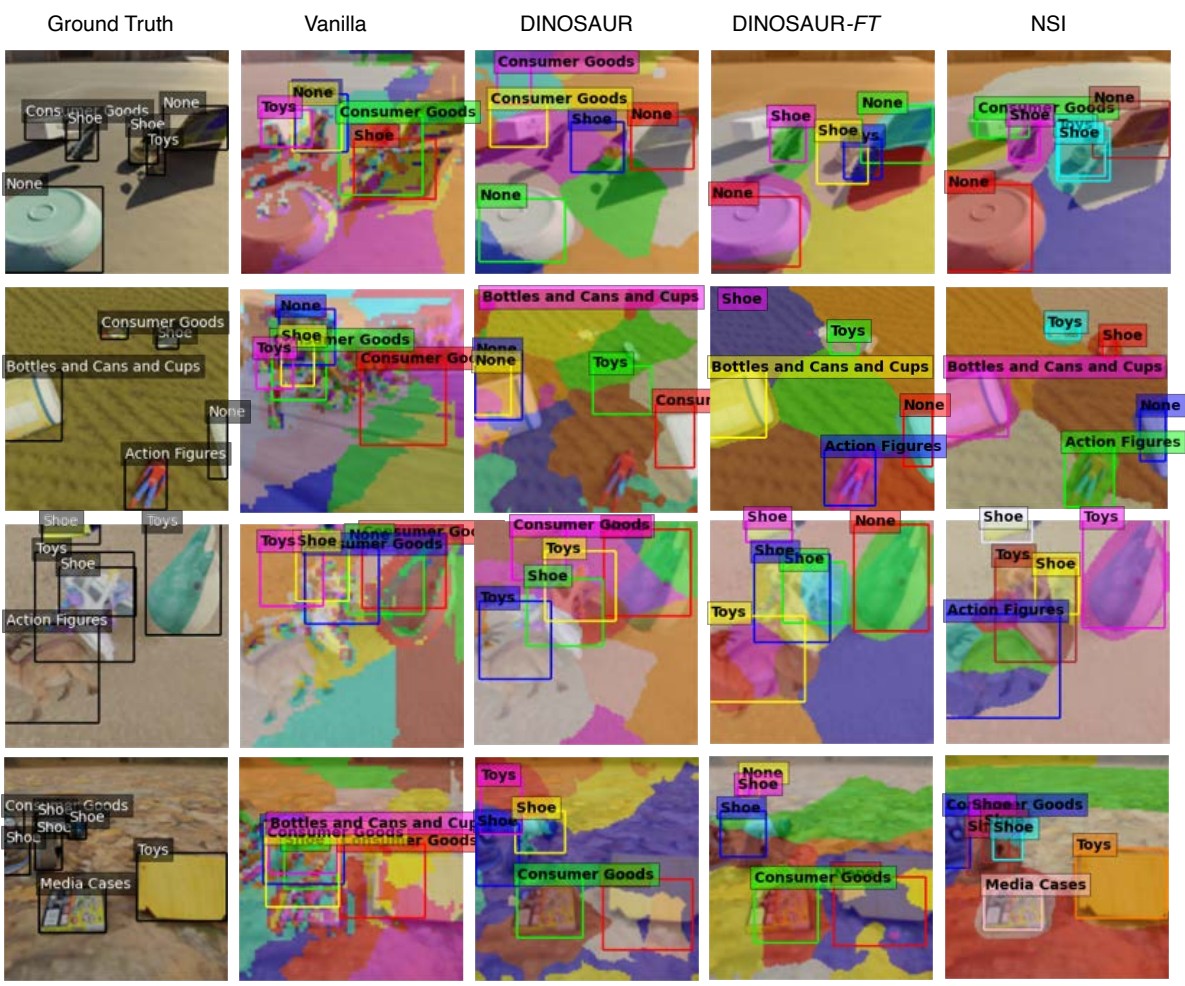

Figure 41: Object detection results on MOVi-C scenes.

