# OpenReview forum: "Neural Slot Interpreters: Grounding Object Semantics in Emergent Slot Representations"
_TMLR — Accepted by TMLR_

### Review · Reviewer_3va7 · 2025-03-19

**Summary Of Contributions:**

This paper introduces Neural Slot Interpreters (NSI), a method to ground semantic information to objects learned by slot attention. The semantic information is represented visXML, a markup syntax for object categories and properties. The grounding is achieved by a structured contrastive loss, which projects slot features and visXML text into a shared latent space and aligns them. NSI is evaluated on multiple tasks, including scene-schema alignment, weakly-supervised visual grounding, object discovery, and learned visual token. The Proposed NSI shows better performance than the compared baselines.

**Audience:**

Yes

**Broader Impact Concerns:**

I see no outstanding concerns about ethical implications.

**Claims And Evidence:**

No

**Requested Changes:**

* Please address my questions in the "Major weaknesses" section, which I think are cirtical to securing my recommendation.
* I also strongly recommend a re-write of the experiment section, clarifying the input/output of each experiment and how exactly each baseline is used in each experiment. Currently, they look crumbled and are not good at conveying the information.
* Please also address the minor weaknesses I mentioned above, which I think will strength the work in my view.

**Strengths And Weaknesses:**

Strengths:

* NSI proposes to adapt the DINO features aggregated by slot attention and align them with visXML text, in a compositional manner. This is interesting.
* The paper provides abundant evaluation of the proposed method on various downstream tasks, showing different aspects and applications of the proposed methods.
* This paper provides many technical details and qualitative results. They help understand the experiment setup and model performance.


Major weaknesses:

* NSI poses a strong assumption that a correct schema that corresponds to the scene exists and is given. This limits the usage of the proposed method compared to the common-used (either open-set or close-set) object detection or segmentation models, which can generate the set of objects for each input image without any assumptions. In such a case, how is NSI useful in practical computer vision tasks? Or, **in what case should I use NSI rather than an object detection(-segmentation)-featurization-captioning pipeline with pretrained models?** (e.g. https://github.com/ttengwang/Caption-Anything)
* Following the above point, the "Scene-Schema Alignment Evaluation" task looks very *artificial* to me. It requires you to have a set of scenes and their corresponding schemas and requires you to shuffle them to construct such a task. I cannot see how it's useful in practice.
* The central question of this paper is “Can slot representations, when explicitly grounded in object semantics, serve as effective compositional abstractions for grounding and reasoning?". Yet I doubt whether this paper addresses this question. Because the slot attention representations are essentially the weighted mean of the underlying featurizer (DINO in this paper). In this regard, this paper shows that DINO features are very good at distinguishing different objects, and taking the mean of them is still semantically meaningful. I wonder how relevant is this to "slot representations". For example, how about simply pooling DINO features according to object detection bounding boxes/masks and aligning them with text?
* How does NSI ground "primitives" into "slots"? The contrastive loss in Section 4.2.4 only explains how the "schemas" are grounded into "scenes". I was expecting another contrastive loss between primitives and slots (as depicted in Figure 3) but it's not mentioned at all in the paper. Without this, how could NSI possibly learn to relate primitives with slots?
* While the paper compares to the CLIP model and claims the advantage of NSI in compositionality, I think a VLM baseline is needed. While the CLIP encoder has "bag-of-word" behavior, some of the recent VLMs do not and have a better performance in understanding texts structured in the visXML-like format.

Minor weaknesses and questions:

* This paper uses a lot of $\Rightarrow$ in sequences of equations that do not actually involve "imply" relationships (Eq. 4, 5, 6, 8, 9). Please use a comma or semicolon when they're simply sequential operations on the input variables.
* I'm not sure why and how visXML is important for this work. Seems that visXML can be replaced by any dictionary-like data structure and it does not affect the method or its performance. If this is the case, I suggest to hide/omit such implementation details in the paper.
* I wonder how NSI handles "background slots". As exemplified in Figure 5(d), the background (floor) also takes up two slots (cyan and gray), but seems somehow they are not matched to any primitives. How is NSI able to do this?
* In Figure 5(f), what do two repeating "<category>person</category>" tags mean?
* In Figure 7, the top row, 3rd image, seems the slots are not accurately aligned with the object. Why is the bounding box still accurate?
* The writing of the paper could be improved. I found myself lost multiple times and struggled to understand when reading this paper.
* In Section 3.1, "N features" -> "L features".
* Figures 14 and 15 are informative. I think it's worth to move them to the main text.

---

> ### Author Response · Authors · 2025-04-05
> **Author Response (1/n)**
>
> We thank the reviewer for their constructive feedback. We address the concerns next:
>
> > Weakness #1 : NSI poses a strong assumption that a correct schema that corresponds to the scene exists and is given. This limits the usage of the proposed method compared to the ......
>
>
> This is a great question! While we agree NSI assumes a known schema, differing from assumption-free detectors, this positions NSI as a powerful tool for property-level understanding in closed-set domains where relevant schemas can be defined. NSI excels at grounding fine-grained object properties (like shape, material, texture) that go beyond simple category labels. Practical examples where this may be beneficial include:
>
> - Creative design tools (when visual asset properties are a priori known).
> - Robotics in controlled settings (e.g., guiding manipulation based on object shape, size, and material).
> - Specific database searches (retrieving scenes based on precise object property combinations).
>
> Notably, unlike traditional pipelines that might require segmentation maps, NSI's grounding process does not require segmentation supervision. However, we concur with the reviewer that NSI is not intended as a universal replacement for general-purpose object detectors.
>
> >Weakness #2: The "Scene-Schema Alignment Evaluation" task looks very artificial to me. It requires you to have a set of scenes .....
>
> We acknowledge the reviewer's concern. However, we believe this task, though constructed, provides a direct and quantifiable means to address the central question guiding our work: whether learned slot representations extracted from scenes can be effectively grounded in object semantics (properties described via schemas). While the setup involving shuffling scenes and schemas creates the task structure, its fundamental purpose is to rigorously test and quantify how well these learned slot representations align with their corresponding property descriptions. Regarding practical utility, as we outlined in the previous reply, the ability of NSI to ground fine-grained object properties lends itself to several potential real-life applications that can be promising future directions motivated by slot based grounding.
>
> > Weakness #3 : The central question of this paper is “Can slot representations, when explicitly grounded in object semantics, serve as ........
>
> First, we would like to clarify that the slot attention mechanism is an iterative rollout of two steps: (1) feature aggregation into slots and (2) updating slots through a recurrent unit. Earlier, we had only mentioned step (1) in the preliminary section and apologize for the confusion. Both these steps are indeed crucial as the iterative refinement allows slots to effectively bind to fine-grained object features (extensive ablations on the slot attention can be found in [1]). To directly address the reviewer's suggestion about simply pooling features based on bounding boxes -- we directly pooled he DINO features within ground truth bounding boxes to represent objects for our  BBox QKV comparison in Section 5.2.2. Our experiments show that this bounding box-based pooling approach (BBox QKV) is less effective at grounding object semantics compared to NSI, which utilizes the full iterative slot attention mechanism. We argue this is because bounding boxes can be coarse and struggle with overlapping objects, whereas the iterative slot attention mechanism encourages the emergence of finer-grained, non-overlapping spatial representations (slots) that better align with object features.
>
>
> >Weakness #4: How does NSI ground "primitives" into "slots"? The contrastive loss in Section 4.2.4 only explains how the "schemas" are grounded into "scenes". ....
>
> This is an insightful question. While a direct contrastive loss between ground-truth primitive-slot pairs could be theoretically optimal, such correpsondences are unknown. NSI addresses this by employing an indirect grounding strategy embedded within the scene-schema contrastive learning framework. Specifically, the similarity score between a scene and schema (detailed in Section 4.2.3) is calculated by summing the similarities of each schema primitive to its nearest neighboring slot representation in a shared latent space. Subsequently, the contrastive learning objective (Section 4.2.4) optimizes this aggregated score to maximize the alignment between correct scene-schema pairs. By optimizing this proxy metric, which intrinsically relies on the nearest-neighbor primitive-slot assignments, NSI encourages the slot and primitive representations to emerge such that semantically related primitives are aligned to the appropriate slots. Notably, this approach overcomes a key limitation of prior work reliant on stricter, often one-primitive-per-slot matching templates (HMC), as discussed in the paper.

---

> ### Author Response · Authors · 2025-04-05
> **Author Response (2/n)**
>
> >Weakness #5: While the paper compares to the CLIP model and claims the advantage of NSI in compositionality, I think a VLM baseline ......
>
> Thank you for this suggestion. We have included GroundingDINO  a VLM pre-trained on a large corpus (>2 million image, text pairs) for open-set object detection and vision-language alignment, as a baseline in our revised manuscript (Section 5.2.1). We evaluated its performance alongside CLIP and our NSI method on the bi-modal scene-property retrieval task across multiple datasets. Our findings (Figure 5) show that while the VLM performs comparable to CLIP, particularly well with broad category alignment (e.g., on MS-COCO).  However NSI demonstrates superior performance on tasks requiring granular, object-centric grounding based on specific properties like shape or texture (e.g., on CLEVrTex, MOVi-C). Even though the Grounding DINO has compositionality, grounding in text seems to be underprovisioned for fine-grained property grounding for scenes.
>
> >Minor Weakness: VisXML contribution
>
> We agree that the conceptual and engineering contributions are confusing. We have re-written this section with importance on the nested schema structure, removing references to XML or other engineering syntax.
>
> >Minor Weakness: Background slots
>
> Since NSI uses nearest neighbour assignment of primitives to slots, no primitives are attached to semantically unmeaningful background slots. The slots division artifact arises from using mores slots than objects in the scene, causing extra slots to bind on the background.
>
> >Minor Weakness: Figure 5(f) and Figure 7
>
> Figure 5(f) -- the two instances of person in the schema get assigned to the yellow slot that covers both the woman and child. Figure 7: Even though the slot partially covers the bag it still contains the bag primitive since it contains the center point (included in the schema)
>
> >Minor Weakness: Writing and Formatting
>
> Thank you for the suggestions, we have incorporated them in the manuscript and have restructured the experiment section for clarity. We also appreciate the attention to detail and catching the typos.
>
> [1] Object-Centric Learning with Slot Attention, Locatello et al. 2020

---

### Review · Reviewer_MjoJ · 2025-03-23

**Summary Of Contributions:**

The paper proposes Neural Slot Interpreters (NSI) which builds on slot attention and DINOSAUR, which uses DINO as the vision backbone for slot attention and aims to reconstruct the features instead of the raw pixels.  NSI builds on DINOSAUR by using an annotation scheme for objects and their properties, a similarity metric to assign multiple concepts to slots (thus breaking the constraint of one-object-per-slot of prior work), and introduces contrastive learning to ground slots, as well as a hierarchical transformer-based architecture.  A series of experiments are conducted to show the effectiveness of the proposed NSI and different design choices (e.g. using Hungarian matching for slot matching, and different visual encoders).

**Audience:**

Yes

**Broader Impact Concerns:**

No concerns

**Claims And Evidence:**

No

**Requested Changes:**

The reviewer feels that the manuscript needs considerable amount of rewriting for clarity, especially wrt to the introduction, related work and experiments.

1. The introduction needs to be reworked to be clearer with claims / hypotheses be reworked to better match what is shown by the experiments.

     a. There is a disconnect because the questions being asked and text that follows: *Can slot representations, when explicitly grounded in object semantics, serve as effective compositional abstractions for grounding and reasoning*

     - The reviewer expected this to be followed by a description of a set of experiments to answer this question, but instead there is a description of the proposed method.  A logical flow to this reviewer would be to describe the proposed method, which key challenges inspired the key modifications to the method, and then start to pose the questions which will be answered by experiments.

    b. In the introduction, the manuscript poses the following questions:

      - *Do NSI modules require specialized training recipes?*
               The reviewer is uncertain what is meant by a “specialized training recipe”.  The manuscript indicates that the “NSI is a plug-and-play paradigm”.  The reviewer does not find the question and statement to be logically connected as it is possible for a framework to be plug-and-play, but specific training recipes to be necessary for the framework to be effective (e.g. some models need to be trained before other parts, or parts need to be pretrained, or certain training strategies - optimizers, learning rate scheduler need to be applied).  Also, the reviewer believes that a specific training recipe is followed for the set of experiments provided.

      - *Are notions of objects effectively grounded in emergent slot representations?* The paper states that “NSI surpasses the state of the art” in section 5.2.1  Due to the limited discussion of prior work, it is not clear to the reviewer whether NSI indeed surpasses the state-of-the-art methods, and what the appropriate set of methods would be considered for this particular setting.  Section 5.2.1 does compare different ways of matching and different visual encoders and shows the effectiveness of the proposed approach.
      -  *Does NSI preserve compositionality?* (Section 5.3) The reviewer does not find competitiveness in object discovery to be necessarily a sign of compositionality, as it is possible to discover objects without notions of compositionality.
      -  *Are NSI-grounded slots effective substrates for visual reasoning?* The experiments in section 5.4 also does not demonstrate visual reasoning.  The setup of the Clevr-Hans experiments are not clearly specified, but it does not appear to follow the reasoning that is needed for the question answering that was designed for the original Clevr dataset.   Instead, the task appears to be a classification task with posthoc explanations.

   c. The presentation of visXML as a contribution feels a bit odd.  While it is reasonable to have well-defined object properties, the more important part is what object properties are vs the specific format (e.g. XML) used to store the properties.  To the reviewer, it seems more appropriate to discuss the concepts vs the specifics of the XML tags.

2. Related work need to have better coverage of prior work on slot attention and object centric learning
- The relation of this work to recent works on improved slot attention for grounding and compositionality [1,2] is unclear
- As there as been prior work that used contrastive learning for object-centric learning [3,4] and different ways to match features and slots [5] it would be good to discuss how these work relate more.
- The relation of this work to program induction is not clear to the reviewer.  It seems unnecessary to include program induction in the related work.  The reviewer would recommend a more thorough review of prior work on using slot attention and object-centric learning.

3. Improved description of the method

- Overall a bit more introduction of slot attention and DINOSAUR would be helpful, and a more clear explanation of how this work builds on DINOSAUR.

- Section 3.3 - this section is missing the most relevant piece of information - i.e. that HMC is typically the main method that prior work uses for matching objects to slots.

4. Introduce the experiments setup for each experiment so the reader can have a clearer understanding of each task.

5. Several of the figures should be improved so that the colors are more distinct and easier to interpret.

   a. Figure 1 - The colors used for "Set Match Slots" and "NSI Slots" are extremely similar.  More distinct colors are recommended.

   b. Figure 5 - Use more distinct colors for different methods (e.g. CLIP and NSI), with same color family for related methods (e.g. NSI variants).

   c. Figure 5,6 - Indicate in caption that higher saturation is R@1, lighter saturation is R@5.

   d. Figure 5, it is difficult to tell the what material the objects are in the picture

6. Some of the equations hard to follow

- Eq. 1: K is used for both the number of slots and the key matrix.
- Eq. 1: It is not clear how the notation K(.), Q(.), V(.) should be interpreted - should K,Q,V be interpreted as matrices or functions?
- Eq. 1: D should be specified as the dimension of the key, query, value.

In general, a brief description in text of what all the symbols are and how the equations are related would be helpful

6. Writing should be improved to be more concise and clearer

- "a similarity metric that explicitly reasons" => It not clear that a similarity metric is capable of reasoning

- Section 3.1 - the terminology "semantic object files" is not introduced.

For simplicity, the reviewer recommend avoiding using this terminology.  Potentially by rewording  the sentence to

"Object-centric learning frameworks decompose scenes by organizing them into semantic object files called slots" => "Object-centric learning frameworks decompose scenes by using \textit{slots} to organize and represent semantic objects."

Alternatively, the authors can properly introduce the concept of semantic object files - this is perhaps in reference to the object files introduced by Kahneman et al [6]?

**References**

[1] Grounded Object-Centric Learning. [Kori et al. 2024]

[2] Identifiable Object-Centric Representation Learning via Probabilistic Slot Attention. [Kori et al. 2024]

[3] Contrastive Training of Complex-Valued Autoencoders for Object Discovery [Stanic et al. 2023]

[4] Towards Self-Supervised Learning of Global and Object-Centric Representations [Baldassarre and Azizpour 2022]

[5] Unlocking Slot Attention by Changing Optimal Transport Costs [Zhang et al. 2023]

[6] The reviewing of object files: Object-specific integration of information. [Kahneman et al. 1992]

**Strengths And Weaknesses:**

**Strengths**
- The ability to discovery and ground objects is an important direction
- The proposed method is interesting and shown to be effective in experiments

**Weaknesses**
- It was difficult to get the main contributions of the work and how it related to prior work on slot attention and object-centric learning
- The claims / questions in the introduction are not properly substantiated by the experiments
- The writing and presentation of the work need to be improved considerably so that the contributions, method, and experimental setup are clear and self-contained (without the reader having to refer repeatedly to prior work)
- Even though the work is following up on a line of work that aims to perform object discovery, it appears to require training.  While there are experiments comparing against weakly supervised visual grounding, it is not clear whether it should also compare against fully supervised visual grounding methods.
- For experiments, it is unclear whether suitable baselines are included.  For instance, for weakly supervised visual grounding methods, perhaps it would make sense to compare against more recent methods such as [1] or zero-shot methods such as [2][3]. The reviewer feels that the direction of this work doesn’t necessarily necessitate such comparisons, but if the manuscript claims to have compared against a state-of-the-art weakly supervised visual grounding method, then it is important that it does so.

[1] Weakly Supervised Open-Vocabulary Object Detection [Lin et al. 2024]

[2] Grounding DINO: Marrying DINO with Grounded Pre-Training for Open-Set Object Detection [Liu et al. 2023]

[3] GroundVLP: Harnessing Zero-shot Visual Grounding from Vision-Language Pre-training and Open-Vocabulary Object Detection [Shen et al. 2023]

---

> ### Author Response · Authors · 2025-04-05
> **Author Response (1/n)**
>
> We thank the reviewer for constructive feedback and attention to detail. We address concerns and questions below.
>
> > Weakness #1 : Even though the work is following up on a line of work that aims to perform object discovery, it appears to require training. While there are experiments comparing against weakly supervised visual grounding...
>
> We acknowledge that NSI involves a training phase to ground object semantics into the learned slots. As noted in the paper, NSI is designed to adapt strong, pre-trained object-centric backbones like DINOSAUR for semantic grounding. Regarding the comparison baseline, we chose weakly supervised visual grounding methods because NSI's supervision relies on scene-level annotations consisting of object property labels organized in a schema, without pre-specified segmentation masks being directly mapped to individual slots during training. This setup aligns more closely with the principles of weakly supervised learning, where grounding emerges from higher-level associations rather than explicit, direct correspondences.
>
> > Weakness #2 : For experiments, it is unclear whether suitable baselines are included. For instance, for weakly supervised visual grounding methods, perhaps it would make sense to compare against more recent methods such as [1] or zero-shot methods such as ....
>
> These are great suggestions and fair comparisons since they also use weak supervision to ground scenes into text. In our grounding experiments (Section 5.2), we have now included GroundingDINO as a key VLM baseline. We selected it because it's trained on a large corpus for visual grounding, and importantly, it utilizes the same DINO encoder backbone as NSI's feature extractor. This allows a direct comparison to assess NSI's slot-based grounding approach against VLM-style grounding using similar visual features. GroundingDINO is particularly effective in aligning broader semantic categories in MS-COCO, likely benefiting from its large-scale pre-training. However,  GroundingDINO underperformed NSI significantly on tasks requiring granular property retrieval (like specific shapes, textures, or scales on CLEVrTex and MOVi-C). This suggests that while powerful, its grounding mechanism, primarily learned through natural language supervision, is less adept at fine-grained, object-centric compositional understanding compared to NSI's explicit modeling via slots and structured schema alignment.
>
> > Requested Change #1 (a): The introduction needs to be reworked to be clearer with claims / hypotheses be reworked to better match what is shown by the experiments...
>
> Thank you for your feedback. We have revised the central hypothesis to the paper and clarify the meaning of grounding in that context before introducing our methods. We welcome any further specific suggestions.
>
> > Requested Change #1 (b): Introduction claims
>
> Thank your for the insights and we have modified the claims accordingly.
> - *"Do NSI modules require specialized training recipes?"* Our intention with the "plug-and-play" description was to emphasize NSI's adaptability, particularly its ability to be effectively integrated with existing, pre-trained object-centric backbones like DINOSAUR, rather than requiring end-to-end training from scratch. We have revised the manuscript to state this point more clearly.
> - *"Are notions of objects effectively grounded in emergent slot representations?"* We chose the bi-modal retrieval task as we believe it offers a direct and quantifiable method for evaluating how effectively object semantics are grounded within the learned slot representations. To the best of our knowledge, directly evaluating this type of fine-grained property grounding using a large-scale retrieval benchmark has not been the focus of prior slot-based grounding work. Our original phrasing, claiming NSI "surpasses the state of the art," was intended to convey that NSI significantly outperformed the relevant baselines we implemented and evaluated within this specific retrieval framework. We have revised the manuscript to state specifically that NSI outperforms these benchmarked methods on our proposed grounding evaluation task.
>
> - *"Does NSI preserve compositionality?"* We agree that conflating compositionality with object discovery can be confusing. We have rewritten this claim to directly reflect our enhancement in object discovery performance.
>
> - *"Are NSI-grounded slots effective substrates for visual reasoning?"* We agree our use of "visual reasoning" for the CLEVr-Hans classification task was imprecise. While the task requires deconfounding attributes, it isn't explicit reasoning. We've corrected the terminology throughout the manuscript to accurately reflect it as a classification task.
>
> > Requested change 1(c): VisXML
>
> We have modified the manuscript to reflect the conceptual heirrachy of the nested schema and defer the engineer details to the Appendix.

---

> ### Author Response · Authors · 2025-04-05
> **Author response (2/n)**
>
> > Requested change #2: Related Work
>
> Thank you for highlighting these works and we have incorporated them into the related works section. [1] learns quantized dictionaries to make prediction from slots in the HMC framework. [3,4] use contrastive learning primarily for unsupervised object discovery or symmetry-breaking [2,5] focus on improving the core slot attention mechanism itself. On the other hand, NSI utilizes contrastive learning for explicit, annotation-driven semantic grounding. Furthermore, techniques focusing on slot identifiability or improved slot mechanisms [2-5], could potentially be integrated within the NSI framework in future work to enhance alignment.
>
> > Requested Change #3,4,5,6: Writing, Formatting and Presentation
>
> We appreciate the attention to detail and have incorporated the reviewer's suggestions. We encourage the reviewer to parse the updated manuscript and we welcome further suggestions. Point 5 (d) -- upon publication of this manuscript, we will release code/demo for readers to understand the texture assets and their alignment with the NSI slots.

---

### Review · Reviewer_SoPX · 2025-03-30

**Summary Of Contributions:**

This paper introduces Neural Slot Interpreter (NSI), which combines Slot Attention with weak supervision in the form of object-level labels to ground object semantics in slots. The supervision is presented in an XML-like schema called visXML and the model is co-trained with the traditional Slot Attention objective and a newly introduced contrastive learning objective. In experiments, the authors show the advantages of their proposed approach when compared to traditional Slot Attention, using Hungarian Matching to ground the slots, as well as several other baselines. Specifically, NSI is shown to effectively ground object object semantics in slots, enhance object discovery, and effectively tokenize the image for the CLEVR-Hans reasoning benchmark.

**Audience:**

Yes

**Claims And Evidence:**

Yes

**Requested Changes:**

Addressing the weaknesses / questions in the above section could strengthen the work, but are not critical to securing my recommendation for acceptance.

**Strengths And Weaknesses:**

**Strengths:**

- The paper is generally well-written and the proposed method was explained clearly.
- The combination of slot attention co-trained with object-level labels is relatively underexplored from my understanding, so this work presents findings not found in previous work.
- The experiments are generally well-done and show the advantages of NSI, especially when compared to the previously used HMC.


**Weaknesses / Questions:**

- In Section 5.2.1, it would be informative to include the standard property prediction task results as comparison.
- For the property retrieval task, what is the breakdown of the task by property type? For Section 5.2.2, it seems that maybe BBox QKV would perform well retrieving the location of the object, but not other properties?
- Related to the previous question, what happens in the case where the object labels do not fully capture the objects? For example, in the CLEVR dataset, what if we only had labels for object color, but not object shape? If we train our model with these labels, do we still effectively capture shape in the slots? Since we still have the Slot Attention reconstructive objective, I imagine that shape will still be captured, but how does it compare to training without the contrastive objective? This is relevant to more complex scenes because the labels may not completely capture the objects and all properties of objects in the scene. We would ideally still want to capture concepts that are not present in the labels.
- Does it make sense to add GroupVIT [1] as a baseline to some of the experiments? They similarly train with a contrastive objective, but use natural language instead of the structured object labels used in this work.
- Are the results in Figure 9(d) cherry-picked? Is it possible to quantify how well the attention maps capture the contents of the task across the entire evaluation dataset? If not, I would soften the language about interpretability related to this figure.
- I do not see the value in presenting the object annotations with XML. It seems this could be represented with any other nested structure. In fact, I feel the introduction of visXML actually adds a bit of confusion and may be more appropriate as an implementation detail in the appendix.


[1] GroupViT: Semantic Segmentation Emerges from Text Supervision. https://arxiv.org/abs/2202.11094

---

> ### Author Response · Authors · 2025-04-05
> **Author Response**
>
> We thank the reviewer for critical feedback and address concerns and questions below.
>
> > Weakness # 1: In Section 5.2.1, it would be informative to include the standard property prediction task results as comparison.
>
> Thanks for the suggestion. We have considered prediction tasks in Appendix E where we evaluated NSI slots and HMC learned slots on object detection. Note that while HMC slots can be directly supervised on the task to predict one object per slot, prediction with a classifier head on NSI slots is tricky since slots can represent more than one object. We trained a transformer based set predictor on NSI correspondences to simulataneouly predict object proeprties and position. The results demonstrate that NSI, which handles multiple objects per slot, can detect many objects with fewer slots whereas HMC struggles as scene complexity increases.
>
> > Weakness # 2: For the property retrieval task, what is the breakdown of the task by property type? For Section 5.2.2, it seems that maybe BBox QKV would perform well retrieving the location of the object, but not other properties?
>
> This is a great question and we ran this experiment.  We  trained and evaluated two separate instances of the retrieval task, one with schema containing only positional labels and other containing only non-positional property labels on MOVi-C. We tested both BBox QKV and NSI on the task and the results are shown in Appendix D1.7. First, we found that retrieval results across both models and both instances are worse off compared to the completely populated schemas. However we observe that BBOx QKV relies on positional properties more than non-positional semantics for retrieval and even outperforms NSI on the former type. On the other hand NSI depends more on non-positional properties and scores significantly higher on retrieving them.
>
> > Weakness # 3: Related to the previous question, what happens in the case where the object labels do not fully capture the objects? For example, in the CLEVR dataset, what if we only had labels for object color, but not object shape? If we train our model with these label....
>
> This is an interesting question. While our current model can't handle incomplete primitives we ran an experiment in Appendix D1.5 where the number of objects in the schema were underspecified with repsect to the number of objects in the scene and we found that NSI is robust to the underspecification compared to other baselines. Our future work involves adding data imputation abilities to the model such that it can also model missing/corrupt labels.
>
> > Weakness # 4: Does it make sense to add GroupVIT [1] as a baseline to some of the experiments? They similarly train with a contrastive objective, but use natural language instead of the structured object labels used in this work.
>
> Thanks for highlighting this work and we have mentioned this work in the related work section. We did not include this as a baseline for two reasons (1) The semantic grouping is emergent and sensitive to the number of grouping blocks and grouping tokens and its not obvious what stack and token size will work for our datasets without searching over different configurations. (2) the contrastive loss proposed in the paper depends on mining hard negative examples via text prompting and its not straightforward to us how this will translate to the schema instances.
>
> > Weakness # 5: Are the results in Figure 9(d) cherry-picked? Is it possible to quantify how well the attention maps capture the contents of the task across the entire evaluation dataset? If not, I would soften the language about interpretability related to this figure
>
> Thanks for the suggestion. While most slot masks in Figure 9 experiments yielded the "correct" visual rationale we did observe a few failure cases and have highlighted them in Appendix Figure 37. We don't know how to quantify them yet and have removed the intepretability claim and presented it as qualitatively useful rationales.
>
> > Weakness # 6: I do not see the value in presenting the object annotations with XML. ...
>
> We apologize for the confusion and have deferred the engineering details to the appendix while highlighting the concept of the nested sub-structure in the main part.

---

### Author Response · Authors · 2025-04-05
**Common Concerns**

We thank all reviewers for raising several great questions, adding detailed comments, and drawing our attention to related works. We have incorporated the feedback in the manuscript and modified parts have been highlighted in blue text.  Below, we address some common concerns and summarize additional results in the manuscript.

### Vision Language Model (VLM) Baseline

We have now included results from GroundingDINO [1] a VLM based on the DINO encoder pretrained for visual grounding on a large corpus. We selected this VLM because it's trained on a large corpus for grounding, and importantly, it utilizes the same DINO encoder backbone as NSI's feature extractor.The updated results can be found in Figure 5/ Section 5.2.1. We found that its performance was weaker than NSI and also other slot-based methods like HMC Matching, on the CLEVrTex and MOVi-C datasets. GroundingDINO performs better when grounding broad categories in MS-COCO. We attribute that the weaker performance on granular properties to the fact that although it's an open-set detector, its grounding in natural language means its training may have emphasized object categories more than specific object properties like its shape, size or textures.

### visXML concern

All reviewers unanimously agreed that visXML is an auxiliary contribution and the hierarchical nature of the schema is more important than the specific syntax used to establish the hierarchy. We have modified the manuscript to reflect the conceptual parts while moving the engineering details to the appendix.

[1] Grounding DINO: Marrying DINO with Grounded Pre-Training for Open-Set Object Detection (Liu et al. 2023)

---

### Decision · Action_Editor_RqkF · 2025-04-28

**Recommendation:** Accept as is

**Comment:**

All reviewers are leaning towards accepting the paper after the revision. The AC agrees with the reviewers and recommends acceptance.

**Audience:**

Yes. The paper is of interest to a reasonably large audience in the TMLR community.

**Claims And Evidence:**

Yes. All three reviewers agree that the submission's claims are supported by accurate, convincing, and clear evidence.